# Hemoglobin in the blood acts as a chemosensory signal via the mouse vomeronasal system

Takuya Osakada[1,2,5], Takayuki Abe[1,2,5], Takumi Itakura[1,2,5], Hiromi Mori[1,2,5], Kentaro K. Ishii [1,2], Ryo Eguchi[1,2], Ken Murata[1,2], Kosuke Saito[1,2], Sachiko Haga-Yamanaka [1], Hiroko Kimoto[1], Yoshihiro Yoshihara[2,3], Kazunari Miyamichi [1,2] & Kazushige Touhara [1,2,4✉]

The vomeronasal system plays an essential role in sensing various environmental chemical cues. Here we show that mice exposed to blood and, consequently, hemoglobin results in the activation of vomeronasal sensory neurons expressing a specific vomeronasal G protein-coupled receptor, Vmn2r88, which is mediated by the interaction site, Gly17, on hemoglobin. The hemoglobin signal reaches the medial amygdala (MeA) in both male and female mice. However, it activates the dorsal part of ventromedial hypothalamus (VMHd) only in lactating female mice. As a result, in lactating mothers, hemoglobin enhances digging and rearing behavior. Manipulation of steroidogenic factor 1 (SF1)-expressing neurons in the VMHd is sufficient to induce the hemoglobin-mediated behaviors. Our results suggest that the oxygen-carrier hemoglobin plays a role as a chemosensory signal, eliciting behavioral responses in mice in a state-dependent fashion.

[1] Department of Applied Biological Chemistry, Graduate School of Agricultural and Life Sciences, The University of Tokyo, Tokyo 113-8657, Japan. [2] ERATO Touhara Chemosensory Signal Project, JST, The University of Tokyo, Tokyo 113-8657, Japan. [3] RIKEN Center for Brain Science, Wako, Saitama 351-0198, Japan. [4] International Research Center for Neurointelligence (WPI-IRCN), The University of Tokyo Institutes for Advanced Study, Tokyo 113-0033, Japan. [5] These authors contributed equally: Takuya Osakada, Takayuki Abe, Takumi Itakura, Hiromi Mori. ✉email: ktouhara@g.ecc.u-tokyo.ac.jp

Animals use the olfactory systems to acquire information about the external environment. Mice, for example, show vigorous sniffing behavior that allows for sensing not only volatile airborne chemicals but also nonvolatile compounds, through direct contact in the nose[1]. Volatile odorants are recognized by odorant receptors in the olfactory epithelium and often elicit acute behavioral responses, such as attraction or aversion[2–5]. In contrast, nonvolatile cues are detected by vomeronasal receptors in the vomeronasal organ (VNO) located beneath the nasal cavity, and usually convey sociosexual information associated with stereotypical behavior or emotional changes[6,7].

Secretions, such as urine, tear fluid, and saliva contain both volatile and nonvolatile olfactory cues. For example, male mouse urine contains major urinary protein 3 (MUP3), which elicits intermale aggression[8]. Male mouse tears contain exocrine gland-secreting peptide 1 (ESP1), which enhances female sexual behavior called lordosis and male aggressiveness[9–12]. Juvenile mice secrete ESP22, which suppresses sexual behavior in adult mice to lessen the number of competitors[13,14]. In addition to these intraspecies signals, the VNO also receives inter-species signals, such as rat cystatin-related protein 1 (ratCRP1), which acts as a pheromone in rats but as a predator signal in mice[15]. Olfactory cues appear to help animals take appropriate reactions to environmental changes by affecting the internal physiological state of the animal.

The VNO expresses two types of G protein-coupled receptors (187 V1Rs and 121 V2Rs) that recognize various environmental cues[16,17]. Only a few vomeronasal ligand-receptor pairs have been revealed; ESP1 and ESP22 are detected by single specific V2R receptors, V2Rp5 (Vmn2r116)[10] and V2Rp4 (Vmn2r115)[14], respectively. The signals received by the specific V2Rs are conveyed to the accessory olfactory bulb (AOB) first, and then to the limbic brain regions such as amygdala and hypothalamus wherein specific neural circuits regulate distinct output behaviors[10,12,14]. The vast majority of V2Rs, however, remain orphan, limiting our understanding of sensory mechanisms underlying VNO-mediated behavioral output.

When we discovered the ESP1 molecule in 2005, we also reported that the submaxillary gland contains a molecule, which activates vomeronasal sensory neurons (VSNs) in the mouse VNO[9]. It was since discovered that the activity was due to contamination of blood in the gland, which motivated us to identify a molecular basis and neural mechanism in both the peripheral and limbic brain regions for the vomeronasal stimulatory activity in the blood. In this study, we first identify a vomeronasal stimulatory molecule in the blood and define its receptor. We then reveal the behavior evoked by the blood factor, and a neural pathway responsible for the output.

## Results

**Mouse VSNs activated by C57BL/6 male mice blood hemoglobin.** We initially investigated a molecule in mice blood, which activate VSNs. When C57BL/6 male mice were exposed to 1 μl of blood, we observed induction of expression of c-Fos, an immediate early gene, or phosphorylation of ribosomal protein S6 (pS6), a sensitive neural activity indicator, in a subset of neurons in the basal layer of the vomeronasal epithelium that expresses $G\alpha_o$ and vomeronasal type 2 receptors (V2Rs) (Fig. 1a, b)[18,19]. Blood-dependent c-Fos expression was also observed in the posterior region of the AOB, the first center of the vomeronasal sensory system where axons of V2R-type neurons project (Fig. 1c)[20]. The response was dose-dependent wherein 3 μl of blood induced the maximal c-Fos response in the mitral/tufted cell layer (M/T) of the AOB (Fig. 1d). These results suggest that blood contains compound(s) that activate V2R-type VSNs.

Since the vomeronasal stimulatory activity was present in the cell lysate, but not in plasma (Fig. 1e), we performed two-step column purification of the cell lysate using ion-exchange DEAE (Fig. 1f) and reverse phase C4 (Fig. 1g) columns and examined c-Fos-induced activity in the AOB. Absorption spectrum and mass spectrometry analyses suggested that the first peak with the absorption of around 400 nm was heme; the second and third peaks were identified as α- and β-globin, respectively (Supplementary Fig. 1). c-Fos induction was only observed in mice stimulated within the peak of β-globin (shown in magenta) (Fig. 1f, g and Supplementary Fig. 1a). Indeed, recombinant β-globin induced c-Fos expression in the VNO and posterior zone of the AOB respectively (Fig. 1h). A dose-dependent electrical response was also observed by purified hemoglobin in the electrovomeronasogram (EVG) recording (Fig. 1i, j). These results demonstrated that the molecule, which induced activation of VSNs, was β-globin in the blood.

**Gly17 on hemoglobin is a crucial interaction site with the receptor.** We then investigated subtype and species specificity of hemoglobin (Hb)-derived activation of VSNs. BALB/c strain mouse blood contains two types of hemoglobin, $Hb^{minor}$ and $Hb^{major}$, that can be separated using DEAE columns; we found that only $Hb^{major}$ had vomeronasal stimulatory activity (Fig. 2a, b and Supplementary Fig. 2a)[21]. C57BL/6 strain mice have only one active form of hemoglobin (Fig. 2a, b). The threshold amount of hemoglobin for activating VSNs was 100–300 μg (Supplementary Fig. 2a), which was equivalent to 1–2 μl of the blood in mice. We also used hemoglobin from many kinds of species as a stimulant to mice to see the difference between species. Hemoglobin from rat, guinea pig and human activated VSNs in mice (Fig. 2b and Supplementary Fig. 2b). Conversely, the hemoglobin from horse showed lower stimulatory activity, and no activity was found in blood from frog or fish (Fig. 2b and Supplementary Fig. 2b). Comparison of the amino acid sequences of hemoglobin among the examined species (Fig. 2c and Supplementary Fig. 2c) revealed that two residues, Gly17 and His78, were changed only in the hemoglobin that showed less activation of VSNs (i.e. Gly17 = > Ala (G17A) in $Hb^{minor}$ and = >Asp (G17D) in horse, His78 = >Asn (H78N) in $Hb^{minor}$) (Fig. 2c). Then, we made two mutant $Hb^{major}$ (G17A and H78N) hemoglobin and checked their activities by c-Fos staining with sections from mice exposed to mutant $Hb^{major}$. The G17A mutant of $Hb^{major}$ lost the activity, while the H78N mutant did not (Fig. 2d, e) (mean ± SEM, control; 8.0 ± 0.76, G17A; 17.4 ± 4.4, H78N; 65.0 ± 19.5, BALB/c Hb; 50.3 ± 9.4), suggesting that Gly17 on the surface of hemoglobin is involved in the interaction with the receptor(s) (Fig. 2f).

**Vmn2r88 is a specific vomeronasal receptor for hemoglobin.** The location of the hemoglobin-activated neurons in the VNO suggests that hemoglobin is recognized by the V2R-type receptor family, which constitutes 121 members in mice[17]. To identify hemoglobin receptor(s), we performed double in situ hybridization using cRNA probes that recognize various V2R clades and an immediate early gene, early growth response protein 1 (Egr1), for the VNO of mice stimulated with hemoglobin[10,14,18]. Egr1-positive cells were co-expressed with cRNA probes that detected all members of the V2Rf clade (Fig. 3a, b), but not other V2R clades (Fig. 3a, b, and Supplementary Fig. 3a). Using more specific probes, we found that Egr1-positive cells were identified by the V2Rf5 probe that detected five V2Rs (Vmn2r88, 89, 121, 122, and 123) of the V2Rf clade (Fig. 3c). Finally, a higher hybridization temperature and a specific probe for Vmn2r88, 89, and 122 (which also detect Vmn2r121 and 123) allowed us to pinpoint Vmn2r88 (Fig. 3c). We next generated Vmn2r88-deficient mice using the

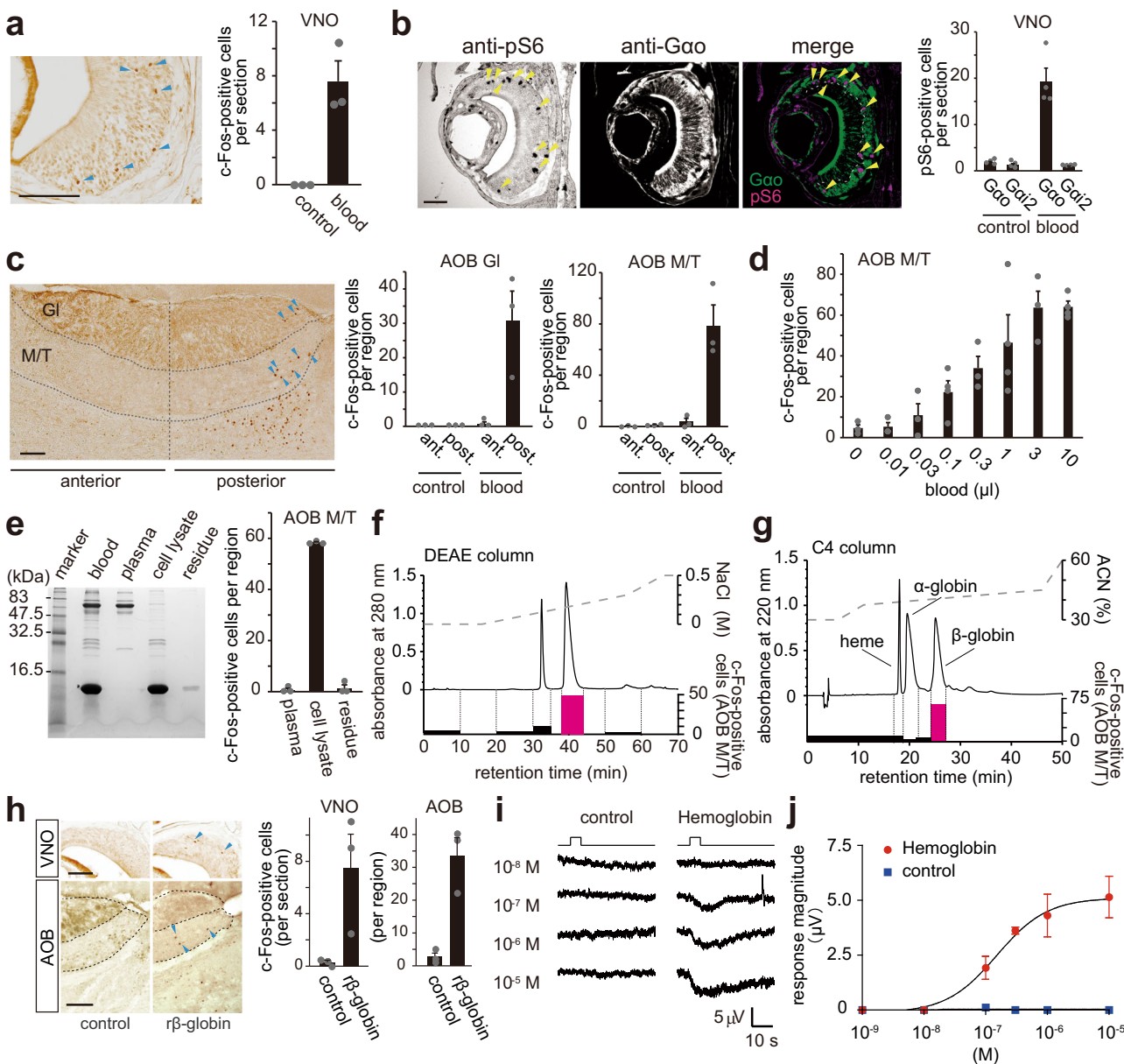

**Fig. 1 Mouse VSNs activated by C57BL/6 male mice blood hemoglobin. a** Representative immunohistochemical image (left) and number of c-Fos-positive cells per VNO section from male mice stimulated by blood (1 μl) or control buffer (right). $n = 3$ for control and blood. Error bars, S.E.M. Arrowheads represent example c-Fos-positive cells. Scale bar, 100 μm. **b** Representative immunohistochemical images of anti-pS6 and anti-Gαo staining (left) and number of total pS6-positive cells in the Gαo and Gα$_{i2}$ zone of the VNO section from blood- or control buffer- stimulated male mice (right). Arrowheads represent example pS6-positive cells (left) and double-positive VSNs (right). $n = 3$ for control and blood. Error bars, S.E.M. Scale bar, 100 μm. **c** Representative immunohistochemical image (left) and number of total c-Fos-positive cells in the glomerular layer (Gl) and mitral/tufted cell layer (M/T) of the AOB section from male mice stimulated by blood (1 μl) (right). Arrowheads represent c-Fos positive cells. $n = 3$ for control and blood. Error bars, S.E.M. Arrowheads represent example c-Fos-positive cells. Scale bar, 100 μm. **d** The number of c-Fos-expressing cells in the M/T cell layer of the AOB sections from male mice stimulated with the indicated amount of blood. $n = 3$ for 0, 0.01, 0.03, 0.3, and 3 μl, and $n = 4$ for 0.1, 1, and 10 μl. Error bars, S.E.M. **e** Separation of blood by centrifuge. Separated blood components: plasma, cell lysate and residue were used for SDS-PAGE analysis and a c-Fos-inducing assay in C57BL/6 male mice. $n = 3$. Error bars, S.E.M. **f**, **g** Two-step HPLC purification with DEAE (**f**) and C4 columns (**g**). Chromatogram and c-Fos-inducing activity of each fraction are shown. The cell lysate fraction was used for DEAE column chromatography and the resultant active fraction was used for C4 column chromatography. The three fraction peaks were defined as heme, α-globin and β-globin by absorption spectrometry and mass spectrometry (Supplementary Fig. 1b-d). **h** c-Fos-inducing activity of recombinant β-globin. $n = 3$. Error bars, S.E.M. Scale bar, 100 μm. Arrowheads highlight example c-Fos-positive cells in the VNO (top) and in the M/T cell layer of the AOB (bottom). **i** Representative EVG recording of hemoglobin-dependent negative change in local field potential of the VNO. **j** Dose-dependent electrical responses of male VNO to hemoglobin. $n = 10$. Error bars, S.E.M.

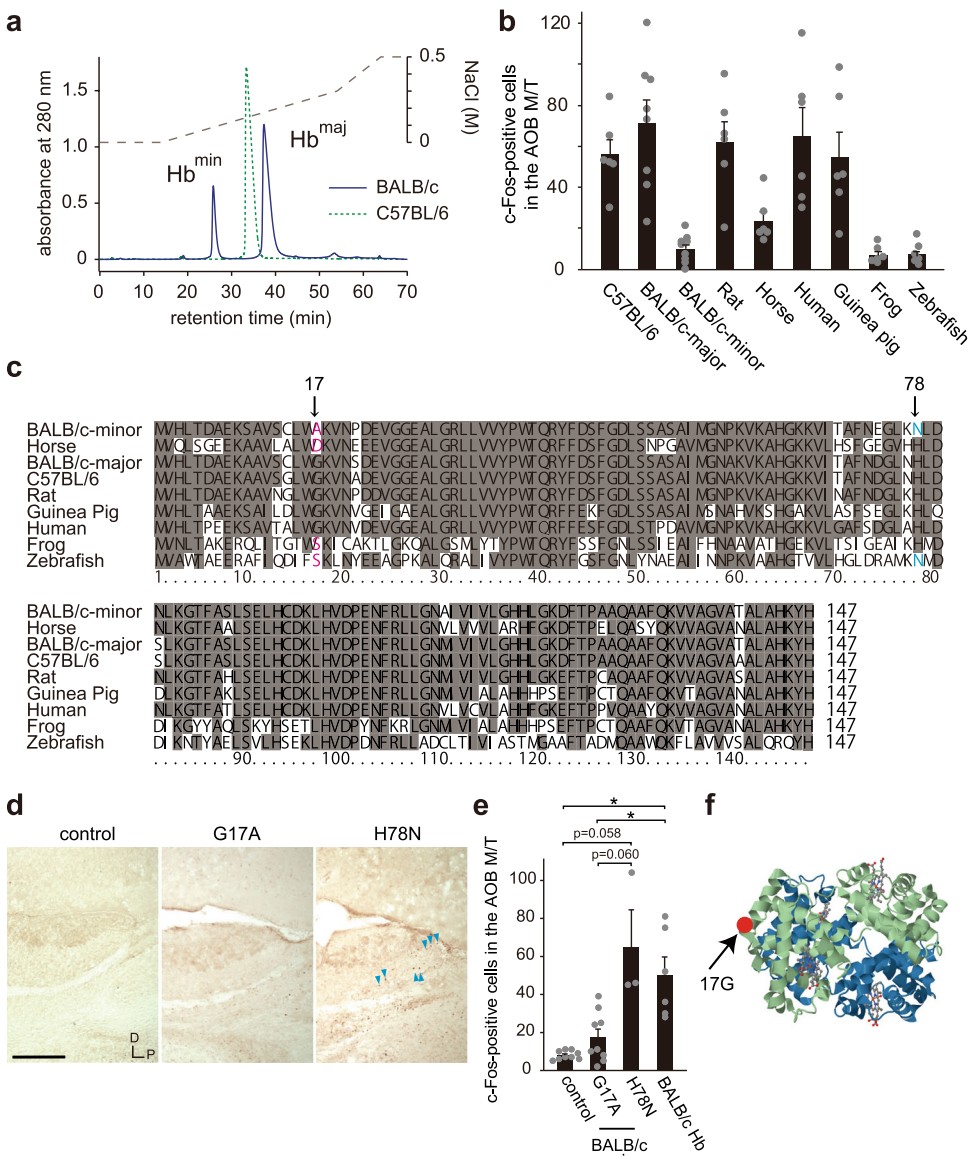

**Fig. 2 Gly17 on hemoglobin acts as a crucial interaction site for ligand-receptor binding. a** Types of hemoglobin in BALB/c and C57BL/6 strain blood cell lysate. Two types of hemoglobin (β-globin) in the BALB/c mouse strain were separated by HPLC with a DEAE column. **b** Number of c-Fos-positive cells in the M/T cell layer of the AOB sections from male mice stimulated with hemoglobin from BALB/c and C57BL/6 male mice blood and blood from various kinds of vertebrates. n = 6 for C57BL/6, Rat, Horse, Human, Guinea pig, Frog, and Zebrafish, n = 8 for BALB/c-major, and n = 9 for BALB/c-minor. Error bars, S.E.M. **c** Alignment of β-globin amino acid sequences. An amino acid G17 is different in BALB/c minor, horse, frog, and zebrafish β-globin. **d** Representative immunohistochemical images of total c-Fos-positive cells in the M/T of the AOB sections from G17A-mutant-, H78N-mutant-, and control buffer-stimulated male mice. n = 3 for H78N, n = 9 for control and G17A. Arrowheads represent example c-Fos-positive cells. Scale bar, 100 μm. **e** Number of c-Fos-inducing cells in the M/T cell layer of the AOB sections from G17A-mutant-, H78N-mutant-, hemoglobin (Hb) from BALB/c-, and control buffer-stimulated male mice. n = 3 for H78N, n = 6 for BALB/c Hb, and n = 9 for control and G17A. Error bars, S.E.M. control vs. BALB/c Hb; p = 0.007, G17A vs. BALB/c Hb; p = 0.046, control vs. H78N; p = 0.058, and G17A vs. H78N; p = 0.060 by the two-sided Steel-Dwass test. **f** Three dimensional structure of human hemoglobin (1A3N, RCSB Protein Data Bank, https://www.rcsb.org/structure/1a3n)[47]. Blue and green cartoons in the model represent α-globin and β-globin, respectively. The position of the 17th glycine is highlighted by a red dot.

CRISPR/Cas9 genome-editing system (Supplementary Fig. 3b)[22] and checked the hemoglobin response (Fig. 3d). Vmn2r88-positive cells were disappeared in *Vmn2r88*-deficient mice (Fig. 3d, e). When mice were exposed to a control vehicle, no cell was activated in mice with (+/+) or without (−/−) Vmn2r88 using pS6 staining, and in the case of hemoglobin exposure, no hemoglobin-dependent activated cell was observed in the VNO sections from *Vmn2r88*-deficient mice (Fig. 3d, e). These histological analyses in the peripheral sensory neurons suggest that hemoglobin is recognized by a single type of V2R, Vmn2r88.

**Hemoglobin enhances *c-Fos* expressing cells in the dorsal region of the VMH and PAG only in lactating mothers.** We investigated whether there was sexual dimorphism or state dependency in the detection of hemoglobin. Both Vmn2r88-expressing neurons in the VNO and cells in the mitral/tufted cell layer of the AOB were activated by hemoglobin not only in males (Fig. 1h and 3e) but virgin and lactating female mice (Supplementary Fig. 4).

To examine which brain regions are activated by hemoglobin, we performed *c-Fos* in situ hybridization mainly targeted for the

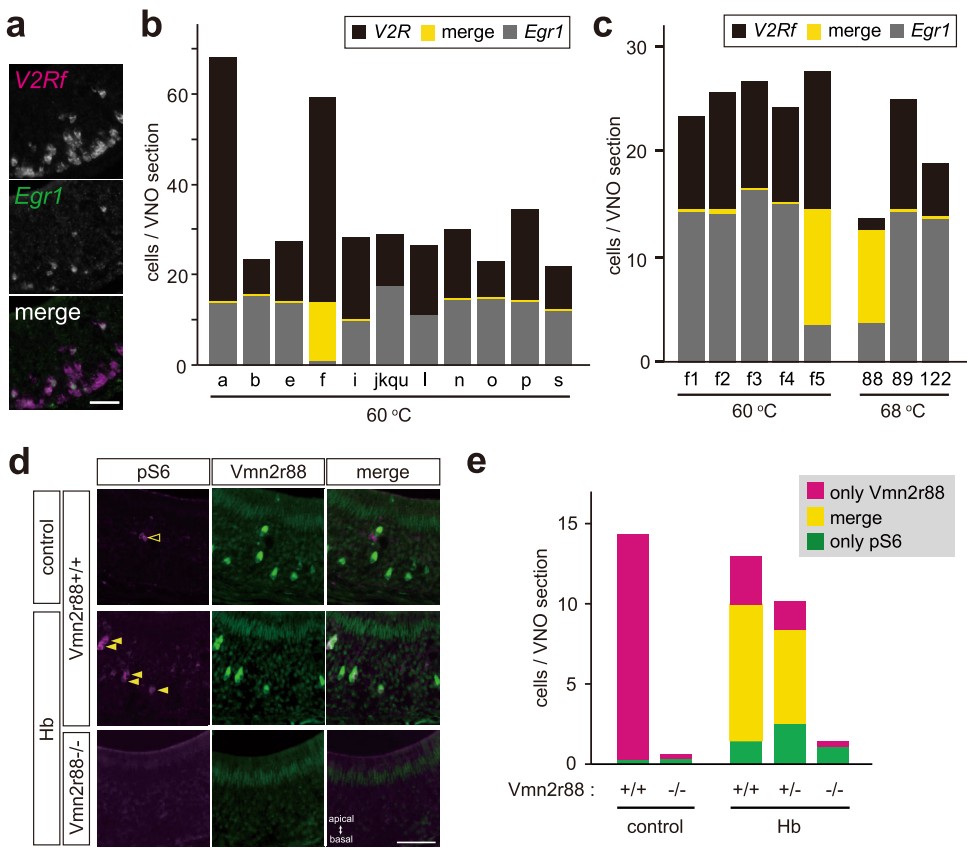

**Fig. 3 Vmn2r88 is a specific vomeronasal receptor for hemoglobin. a** Dual-color ISH staining of a VNO section from a hemoglobin-stimulated C57BL/6 mouse labeled with the *Egr1* cRNA probe (green) and *V2Rf* clade-specific cRNA probe (magenta). *n* = 3. Scale bar, 50 μm. **b**, **c** Bar graph representing the average cell number in VNO sections from C57BL/6 mice stimulated with hemoglobin. In Fig. 3b, probes to distinguish each V2R clade were used. In Fig. 3c, probes to narrow down candidate genes in V2Rf clade (named V2Rf1-V2Rf5) and to further distinguish each gene in V2Rf5 clade (Vmn2r88, 89 and 122) were used. Black, yellow, and gray parts represent only *V2R*-positive (**b**) or *V2Rf*-positive (**c**), *V2R* and *Egr1* double-positive, and only *Egr1*-positive cells respectively. Hybridization temperatures are shown below the graphs. **d** Vmn2r88 and pS6 immunostaining of VNO sections from *Vmn2r88*^+/+ or *Vmn2r88*^−/− male mice exposed to hemoglobin (Hb) or distilled water (control). Open arrowheads show pS6-positive cells; closed arrowheads show cells double-labeled for pS6 and Vmn2r88. In the mutant mice, corresponding pS6 and Vmn2r88 expression completely disappeared. *n* = 3 for *Vmn2r88*^+/+-control, and *Vmn2r88*^-/--Hb, and *n* = 5 for *Vmn2r88*^+/+-Hb. Scale bar, 50 μm. **e** The number of pS6- (green), Vmn2r88- (magenta), and double-positive cells (yellow) per VNO section. In the group of *Vmn2r88*^+/+-Hb 16 sections from each of 5 animals were quantified and 16 sections from each of 3 animals were counted in the other groups. Double-positive cells completely disappeared in the sections from Hb-stimulated *Vmn2r88*^-/- mice.

medial amygdala (MeA), bed nucleus of the stria terminalis (BNST), posteromedial cortical amygdaloid nucleus (PMCo), medial preoptic area (MPA), and VMH, these being regions that receive input from the AOB[23]. Some increase in the number of *c-Fos*-positive neurons was observed in the MeA, mainly the posteroventral region of the MeA (MeApv), in all types of mice tested (males, and virgin and lactating females) upon stimulation with hemoglobin (Fig. 4a, b). Interestingly, we observed a significant increase in the number of *c-Fos*-positive neurons in the dorsal VMH (VMHd) (Fig. 4d, e) and dorsal periaqueductal gray (PAGd) (Fig. 4g, h) only in lactating mothers. No apparent activation was seen in the BNST, PMCo (Fig. 4c), ventrolateral VMH (VMHvl) (Fig. 4d, e), and MPA (Fig. 4f) in any type of mouse. Hemoglobin-dependent *c-Fos* induction was not present in sections from *Vmn2r88*-deficient lactating mice (Supplementary Fig. 5), confirming that the signals are hemoglobin-Vmn2r88-specific. Our histological results showed mother-specific *c-Fos* enhancement in the VMHd and PAGd, suggesting that hemoglobin possesses some specific information for lactating females.

**Hemoglobin enhances digging behavior in lactating mothers.** Mice in their natural environment encounter blood under specific

conditions, such as upon an injury due to intermale aggression, damage by predator attack, and pup delivery. Therefore, we first investigated the effects of hemoglobin on social behaviors such as aggression and sexual behavior. However, there was no obvious change in male-male aggression, maternal aggression, or sexual behavior in virgin females upon hemoglobin exposure (male-male aggression [total events of attack behavior (mean ± SEM)]; cast male intruder; 7.2 ± 2.3, cast male intruder with hemoglobin; 9.5 ± 8.2, *n* = 4, maternal aggression [total events of attack behavior (mean ± SEM)]; cast male intruder; 26.1 ± 3.2, cast male intruder with hemoglobin; 29.8 ± 5.4, *n* = 9, female sexual behavior [total events of lordosis behavior (mean ± SEM)]; control; 0.75 ± 0.75, hemoglobin; 0.67 ± 0.49, *n* = 4 in control, *n* = 6 in hemoglobin). Since the AOB-MeApv-VMHd-PAGd pathway seems to be activated by hemoglobin only in lactating mothers (Fig. 4), and this conceivable pathway is supported by previous studies about circuits responsible for vomeronasal information and hypothalamic neural circuits[12,23,24], we next looked at the behavior of lactating mothers upon exposure to hemoglobin. When a hemoglobin-soaked cotton swab was presented to mothers after pups were removed, the mothers showed robust digging behavior (Fig. 5a, b). The same digging behavior was also observed with exposure to fresh blood (Fig. 5b) (Mother: mean ± SEM, control;

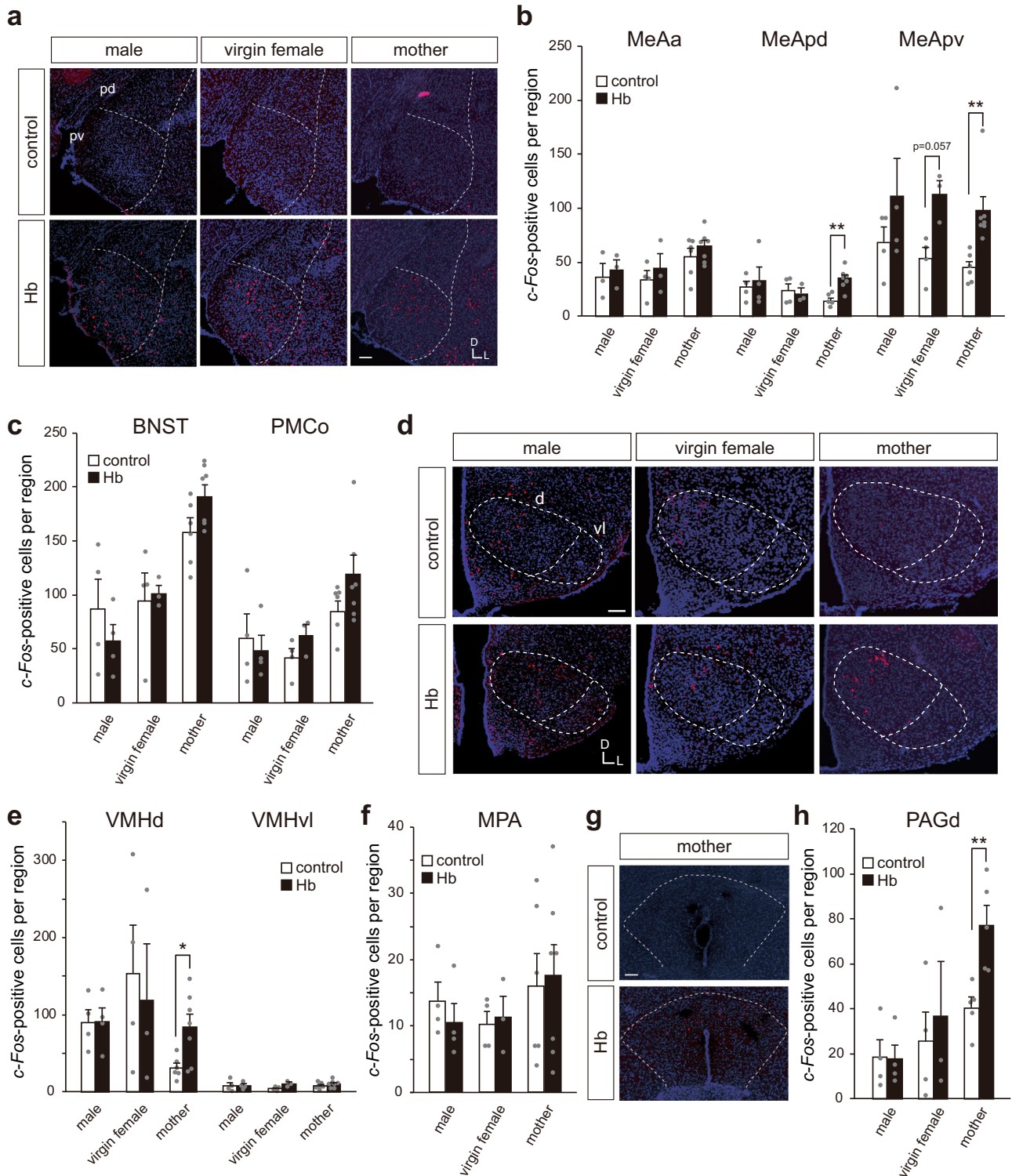

55.1 ± 7.8, hemoglobin (Hb); 111.8 ± 18.1, ESP1; 34.0 ± 14.0, fresh blood; 118.4 ± 17.8). This behavior was not due to pup removal because the same behavior was seen in the trials with mothers together with their pups (Supplementary Fig. 6 and Supplementary Movie 1) (mean ± SEM, digging time, control; 5.5 ± 3.4, hemoglobin (Hb); 22.5 ± 5.4, number of digging, control; 1.4 ± 0.87, hemoglobin (Hb); 9.0 ± 1.9), and consequently, we observed a significant delay in the pup retrieval assay (Supplementary Fig. 7) (mean ± SEM, digging time, control; 164.8 ± 33.5, hemoglobin (Hb); 289.3 ± 38.3). Hemoglobin-dependent digging enhancement

was not observed in *Vmn2r88*-deficient lactating females, suggesting that this behavior is hemoglogin-Vmn2r88-specific (Fig. 5c) (mean ± SEM, +/+ (control); 48.2 ± 5.3, +/+ (hemoglobin); 123.1 ± 19.4, -/- (hemoglobin); 58.2 ± 12.2). In contrast, no increase in digging behavior was seen in a virgin female or male mouse, upon stimulation with hemoglobin (Fig. 5d), consistent with histology data that showed a difference in the activation pattern of brain regions.

Digging behavior may be a reflection of stress-, fear- or anxiety-related emotional changes[25,26]. Thus, we first examined

**Fig. 4 Hemoglobin enhances *c-Fos* expressing cells in the dorsal region of the VMH and PAG only in lactating mothers. a** Representative ISH images of the posterior region of the MeA from C57BL/6 male, virgin female, and lactating mothers pre-stimulated with control- or hemoglobin (Hb)-cotton swabs. *c-Fos* cRNA probe (red) was used in conjunction with nuclear DAPI staining (blue). Abbreviations: MeApd, MeA posterodorsal region; MeApv, MeA posteroventral region; D, dorsal; and V, ventral. $n = 3$ for male-control and Hb, virgin female-Hb, $n = 4$ for virgin female-control, $n = 6$ for mother-control, and $n = 7$ for mother-Hb. Scale bar, 100 μm. **b** The number of *c-Fos*-positive cells in each sub-region of the MeA stimulated with control buffer or Hb. Abbreviations: MeAa, MeA anterior region. For counting MeAa; $n = 3$ for male-control and Hb, virgin female-Hb, $n = 4$ for virgin female-control, $n = 6$ for mother-control, and $n = 7$ for mother-Hb. 6 (MeAa) and 8 (MeAp) sections from each animal were quantified. Error bars, S.E.M. (MeApd) p = 0.004, and (MeApv) virgin female; p = 0.057, and mother; p = 0.001 by the two-sided Wilcoxon rank-sum test. **c** The number of *c-Fos*-positive cells in the BNST and PMCo stimulated with control buffer or Hb. $n = 3$ for virgin female-Hb, $n = 4$ for male-control and Hb, virgin female-control, $n = 6$ for mother-control, and $n = 7$ for mother-Hb. 7 (BNST) and 5 (PMCo) sections from each animal were quantified. Error bars, S.E.M. **d** Representative ISH images of the VMH from males, virgin females, and lactating mothers, stimulated with control buffer or hemoglobin. *c-Fos* cRNA probe (red) was used in conjunction with nuclear DAPI staining (blue). Abbreviations: d, dorsal region; vl, ventrolateral region. $n = 3$ for virgin female-Hb, $n = 4$ for male-control and Hb, virgin female-control, $n = 6$ for mother-control, and $n = 7$ for mother-Hb. Scale bar, 100 μm. **e** Quantification of *c-Fos*-positive neurons in the VMH. The number of sections counted to determine the number of *c-Fos*-positive neurons in each brain area of each animal was 10. Error bars, S.E.M. $n = 3$ for virgin female-Hb, $n = 4$ for male-control and Hb, virgin female-control, $n = 6$ for mother-control, and $n = 7$ for mother-Hb. p = 0.034 by the two-sided Wilcoxon rank-sum test. **f** Number of *c-Fos*-positive cells in the MPA stimulated with control buffer or Hb. $n = 3$ for virgin female-Hb, $n = 4$ for male-control and Hb, virgin female-control, $n = 6$ for mother-control, and $n = 7$ for mother-Hb. 4 sections from each animal were quantified. Error bars, S.E.M. **g** Representative ISH images of the dorsal region of the PAG from lactating mothers stimulated with control buffer or hemoglobin. *c-Fos* cRNA probe (red) was used in conjunction with nuclear DAPI staining (blue). $n = 5$ for mother-control and Hb. Scale bar, 100 μm. **h** Quantification of *c-Fos*-expressing neurons in the PAGd. $n = 3$ for virgin female-Hb, $n = 4$ for male-control and Hb, virgin female-control, $n = 5$ for mother-control and Hb. 4 sections from each animal were quantified. Error bars, S.E.M. p = 0.007 by the two-sided Wilcoxon rank-sum test.

whether 2-methyl-thiazoline (2MT), an odorant that causes innate fear responses, such as freezing, can also elicit digging behavior in mothers[27]. 2MT (100-fold dilution) induced freezing in lactating females as previously described, and as the concentration decreased, the freezing behavior disappeared (Fig. 5e). Conversely, significant increases in digging behavior were observed at the concentrations around which freezing behavior disappeared (5000-fold or 10,000-fold, Fig. 5f). These data sets showed that hemoglobin-induced digging behavior only in mothers and that 2MT at a lower concentration also evoked the same behavioral output. The response towards 2MT exposure suggests that digging enhancement can be interpreted as risk assessment, a mild form of defensive behavior[28].

**Hemoglobin enhances rearing, a type of exploratory behavior, in lactating mothers.** Next, to examine whether hemoglobin produces anxiogenic effects, negative valence, or stress-inducing effects, we performed an open field assay for mothers with or without pre-exposure to hemoglobin (Fig. 6a). Unexpectedly, total distance, center time, and moving speed were the same between all conditions, including hemoglobin, 2MT, and control vehicle (Fig. 6b), suggesting that the experimental paradigm here with pre-exposure of stimulant in the home cage is not suitable to detect anxiety-like behaviors. Instead, we observed a significant increase in the duration of rearing behavior that is a type of exploratory response that can be observed in the open field maze, which was completely abolished in *Vmn2r88*-deficient lactating females (Fig. 6b, c) (mean ± SEM, + / + (control); 57.5 ± 5.1, +/+ (hemoglobin); 76.9 ± 5.7, -/- (hemoglobin); 48.7 ± 4.2)[29,30]. We also performed a simple two-chamber test; lactating female mice apparently avoided the area of 2MT, while mice did not show any avoidance to hemoglobin, suggesting that hemoglobin does not possess negative valence, as 2MT does (Supplementary Fig. 8a-c). A possibility that the behavior is due to a stress response was also examined by measuring corticosterone upon stimulation with hemoglobin, but no apparent increase was observed (mean ± SEM, control; 30.8 ± 10.9, hemoglobin; 38.5 ± 5.3, $n = 5$), suggesting that hemoglobin is not a stressor for the mice. These lines of evidence suggest that the digging and rearing behavior caused by hemoglobin is not an anxiety-related stressful response but a type of exploratory and/or risk assessment behavior.

**SF1-positive cells in the VMHd are important for hemoglobin-dependent digging enhancement.** Finally, we performed experiments to show the importance of specific cell populations for hemoglobin-dependent outputs. Our histological analysis suggests that the activation patterns were different in the cells of the VMHd among lactating, virgin male and female mice (Fig. 4e). To examine whether hemoglobin activates neurons expressing steroidogenic factor 1 (SF1) that are known to be a marker of VMHd, we performed dual-color in situ hybridization with sections from hemoglobin-stimulated lactating females (Supplementary Fig. 9a)[31]. There was significant overlap between *SF1*-expressing neurons and hemoglobin-derived *c-Fos* and its number was larger than that of *c-Fos* and *SF1*-negative cells (the number of *SF1*-negative and Hb-dependent *c-Fos* + cells; 37.8 ± 7.8, the number of *SF1*-positive and Hb-dependent *c-Fos* + cells; 60.3 ± 14.6, both shown in black in Supplementary Fig. 9b). These results suggest that SF1 appears to be a good molecular marker to manipulate hemoglobin-responsive cell population in the VMHd.

Next, we used virus encoding DREADD-Gi and lactating *SF1-Cre* female mice or wild type C57BL/6 female mice as a control to silence neural activities of the SF1-positive cells in the VMHd (Fig. 7a, b and Supplementary Fig. 10a, b). Virus injection into neurons in the VMHd was performed before mating. After waiting for sufficient viral infection and delivery, CNO or saline injection into lactating female mice was performed 60 min prior to hemoglobin exposure and at the same time, their pups were removed from the cage. As a result, neural silencing of SF1-expressing cells in the VMHd significantly suppressed digging behavior in animals that were stimulated with hemoglobin (Fig. 7c and Supplementary Fig. 10c). These results suggest that SF1-positive neurons in the VMHd are necessary for hemoglobin-dependent digging enhancement in lactating females.

We then asked whether the activation of the SF1-positive population induced hemoglobin-dependent behavior. For this purpose, we conducted an optogenetic experiment to activate SF1-positive cells in the VMHd. Optical fibers were implanted above SF1-positive neurons in the VMHd after injection of virus expressing channelrhodopsin (ChR2) in a Cre recombinase-dependent manner (*AAV-DIO-ChR2*) or *AAV-DIO-GFP* virus as a control (Fig. 7d, e)[32]. Weak light stimulation (0.01 mW) on the SF1-positive population elicited a significant increase in the total

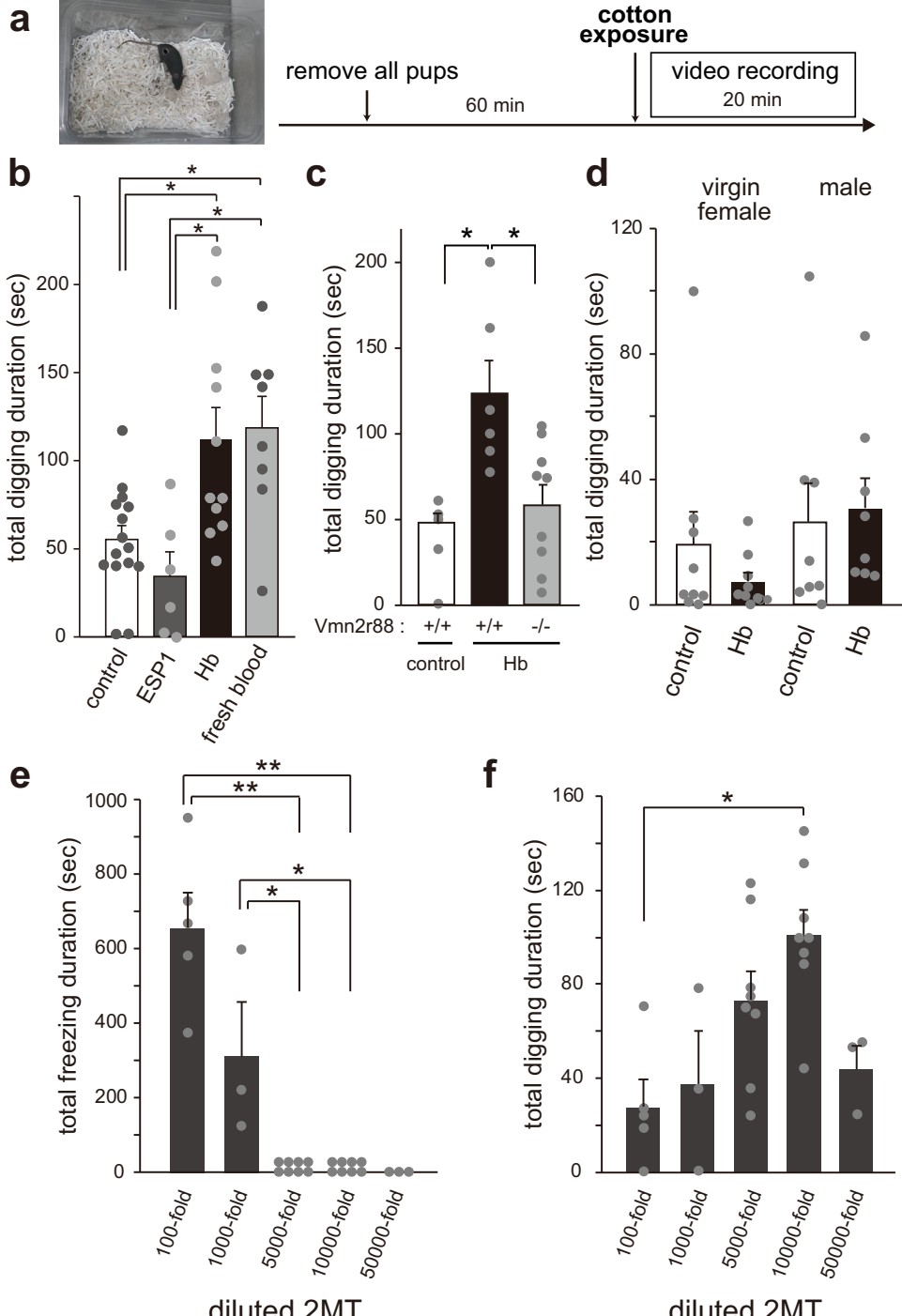

**Fig. 5 Hemoglobin enhances digging behavior in lactating mothers. a** Schematic illustration of the timeline of the cotton exposure assay. In the trials of lactating mothers, their pups were removed one hour before the cotton exposure. **b–d** Quantification of the digging time duration (sec) of wild type C57BL/6 lactating mothers (**b**), *Vmn2r88*-mutant lactating mothers (**c**), males, and virgin females (**d**) with pre-exposure to cotton balls transfused with hemoglobin (Hb), and fresh blood. Wild type C57BL/6 mothers; *n* = 15 for control, *n* = 6 for ESP1, *n* = 11 for Hb, and *n* = 8 for fresh blood, *Vmn2r88*-mutant lactating mothers; *n* = 5–9, virgin females; *n* = 9, males; *n* = 8. Error bars, S.E.M. (**b**) control vs. fresh blood; p = 0.028, control vs. Hb; p = 0.049, ESP1 vs. fresh blood; p = 0.048, and ESP1 vs. Hb; p = 0.044, and (**c**) +/+ (control) vs. +/+ (Hb); p = 0.017 and +/+ (Hb) vs. -/- (Hb); p = 0.041 by the two-sided Steel-Dwass test in panel (**b**) and (**c**). **e, f** Quantification of the total freezing duration (**e**) and digging time duration (**f**) of lactating mothers stimulated with the indicated concentration (from 100-fold to 50000-fold dilution with mineral oil) of a 2MT transfused cotton swab. *n* = 3 for 1000-fold and 50000-fold, *n* = 5 for 100-fold, *n* = 8 for 5000-fold and 10000-fold. Error bars, S.E.M. (**e**) 100-fold vs. 10000-fold; p = 0.007, 100-fold vs. 5000-fold; p = 0.007, 1000-fold vs. 10000-fold; p = 0.015, and 1000-fold vs. 5000-fold; p = 0.015, and (**f**) p = 0.043 by the two-sided Steel-Dwass test.

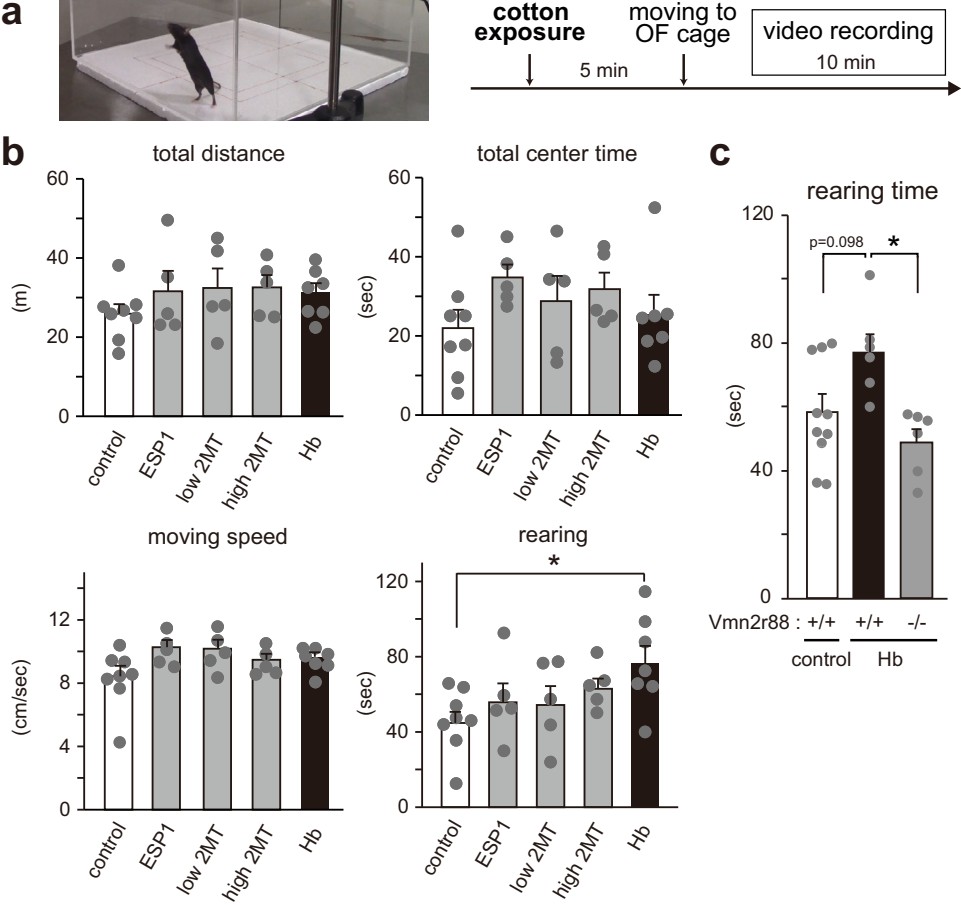

**Fig. 6 Hemoglobin enhances rearing, a type of exploratory behavior, in lactating mothers. a** Schematic illustration of the timeline of the open field (OF) assay with cotton pre-exposure. Cotton exposure was performed in their home cage (with their pups). **b** Quantification of total distance, total center time, moving speed, and rearing time duration of lactating mothers, pre-stimulated with control buffer-, ESP1-, 2MT- (low 2MT: 10000-fold dilution, high 2MT: 10-fold dilution) or hemoglobin (Hb)-cotton swabs in the open field assay for 10 min. $n = 8$ for control, $n = 7$ for Hb, and $n = 5$ for ESP1, low 2MT, and high 2MT. Error bars, S.E.M. $p = 0.048$ by two-sided Wilcoxon rank-sum test with Dunnett correction. **c** Quantification of rearing time duration of *Vmn2r88*-mutant lactating mothers, pre-stimulated with control buffer- or Hb-cotton swabs. $n = 6$ for $+/+$ (Hb) and -/- (Hb), $n = 10$ for $+/+$ (control). Error bars, S.E.M. $+/+$ (control) vs. $+/+$ (Hb); $p = 0.098$ and $+/+$ (Hb) vs. -/- (Hb); $p = 0.011$ by the two-sided Steel-Dwass test.

duration of digging behavior (Fig. 7f, g and Supplementary Fig. 11a–e). In contrast, 1 mW light stimulation did not evoke digging or rearing enhancement but freezing, jump, and dash (Supplementary Fig. 11f, g) (3 out of 5 mice showed freezing after 1 mW light stimulation.). Therefore, SF1-positive population in the VMHd is sufficient for exploratory and/or risk assessment behavior depending on the strength of stimulation.

## Discussion

In this study, besides the main role of hemoglobin as an oxygen carrier in the blood, we discovered a previously unrecognized function of hemoglobin as a chemosensory signal. Recently, hemoglobin was also identified as a vomeronasal-stimulating ligand in the course of analysis of a molecular basis for infanticide[33]. Hemoglobin itself, however, did not induce a pup attack, and *Vmn2r88* knock-out had little effect on behavior[33]. This study revealed the interesting behaviors of digging and rearing in mothers, upon the nasal sensing of hemoglobin. This discriminative output can be interpreted as exploratory and/or risk assessment behavior[28,34]. This hemoglobin-induced behavior appears to be caused by some change in the internal state of lactating females, via activation of specific brain regions such as the VMHd and PAGd (Supplementary Fig. 11h).

What is the meaning of hemoglobin-induced behavior in mice that is only apparent in motherhood? We noticed that without hemoglobin, the duration of digging behavior was longer in mothers than in males or virgin females (Fig. 5b-d), shorter in the presence of pups in the nest than in the absence (Supplementary Fig. 6b), and much longer in the pup retrieval assay (Supplementary Fig. 7c). This difference in duration of background digging behavior may be a reflection of lactating females' susceptibility to their external world and/or exploratory activity towards their surrounding environment, which was enhanced by hemoglobin. The hemoglobin effect may be related to the experience of blood exposure upon birthing. There is also a possibility that hemoglobin-dependent digging is a type of repetitive behavior in mice[35,36].

We showed that hemoglobin activated a limbic neural pathway, AOB-MeApv-VMHd-PAGd, in lactating mothers (Supplementary Fig. 11h). Interestingly, these brain regions are the same as those responsible for ESP1-dependent sexual behavior and also those that respond to signals from predators, such as snake skin[12] and rat CRP1 (Ref[15].). The pathway at the level of a nerve nucleus appears to be the same but responsible cell populations must be distinct from each other as was shown for the case of ESP1, snake skin, and ESP22 (Ref[12,14].). Another interesting point on the circuit is that the behavioral output was observed only in lactating mothers, even though both hemoglobin receptor-expressing cells

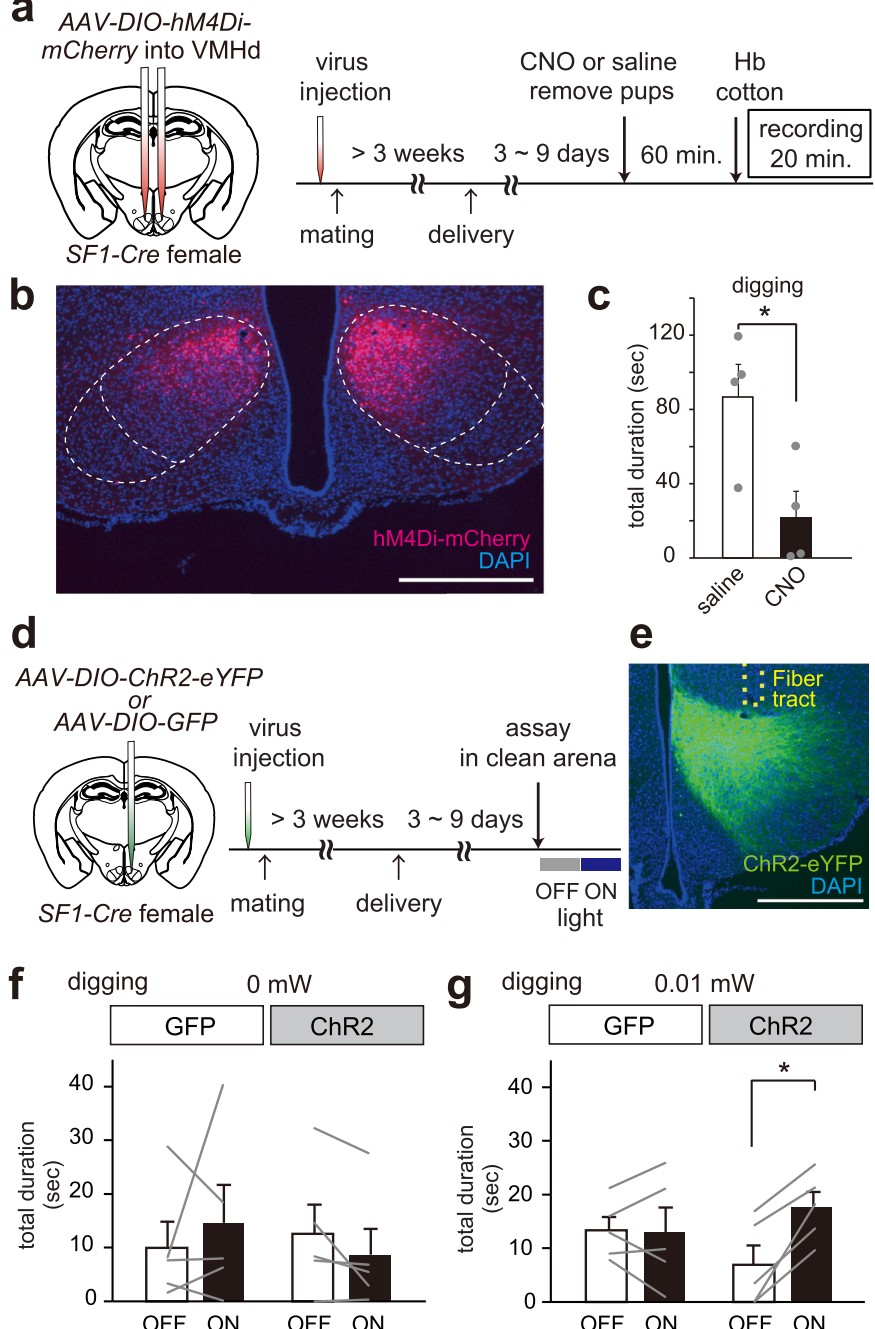

**Fig. 7 SF1-positive cells in the VMHd are important for hemoglobin-dependent digging enhancement in lactating female mice. a** Schematic illustration of the animal setup and timeline for pharmacogenetic inhibition of *SF1*-expressing neurons in the VMHd. *AAV-DIO-hM4Di-mCherry* is injected into SF1-positive cells in the VMHd. Image adapted from Allen Mouse Brain Atlas[48]. **b** A representative coronal section showing DREADD-Gi expression (mCherry-positive cells shown in red) in the VMHd. *n* = 8. Scale bar, 500 μm. **c** Quantification of the total digging duration (sec) of hemoglobin (Hb)-stimulated *SF1-Cre* lactating mothers with i.p. injection of saline or CNO. *n* = 4 for CNO group. *n* = 4 for saline group. Error bars, S.E.M. p = 0.030 by the two-sided Wilcoxon signed-rank test. **d** Schematic image and illustration of the setup and timeline for optogenetic activation of *SF1*-expressing neurons in the VMHd of lactating females. *AAV-DIO-ChR2* or *AAV-DIO-GFP* are injected into SF1-positive cells in the VMHd and optic fibers are implanted above the target region. Image adapted from Allen Mouse Brain Atlas[48]. **e** A representative coronal section showing ChR2 expression (eYFP-positive cells shown in green) in the VMHd. *n* = 5. Scale bar, 500 μm. **f, g** Quantification of digging behavior, with or without weak light stimulation (**f** 0 mW, **g** 0.01 mW). Error bars, S.E.M. p = 0.043 by the two-sided Wilcoxon signed-rank test, *n* = 5 for GFP and ChR2 group.

in the VNO and neurons in the mitral/tufted cell layer of the AOB were similarly activated in male and virgin female mice (Supplementary Fig. 4). These results suggest that the difference in the behavioral response toward hemoglobin between virgin and lactating females is not due to the sensitivity to the ligand at the peripheral level. Furthermore, our results show that the

representation of sensory information becomes distinct between virgin and lactating females in the downstream brain areas (Fig. 4). Future studies to investigate how the animals reproductive state affects the sensory information process in these brain areas are critical to fully understand neural mechanisms underlying the state-dependent behavior modulation.

Although it is known that the volatile odors of bleeding serve as an aversive or attractive cue to animals, including humans[37,38], this study provides evidence that nonvolatile cues in the blood also possess information regarding the external environment. The detection of hemoglobin takes place in the vomeronasal organ at the bottom of the nasal cavity through a single G protein-coupled receptor, and the signal is processed via dedicated limbic regions. Our study demonstrates that sensing the 'smell of blood' occurs in a state-dependent fashion, namely only in birth-experienced lactating females, which may be crucial for protecting pups and ensuring further parental behavior in urgent situations.

## Methods

**Animals**. Animals were housed under a regular 12 h dark/light cycle, 23 ± 2 °C, 50 % humidity, with food and water *ad libitum*. Wild type BALB/c, C57BL/6 mice were purchased from Japan CLEA (Japan), Japan SLC (Japan), or Charles River Japan (Japan) for all experiments. *Vmn2r88*-deficient mice were generated as described in the "Generation of mutant mice by CRISPR-mediated genome editing". SF1-Cre (also known as *Nr5a1-Cre*, Jax#012462) mice were purchased from the Jackson Laboratory. Experiments were carried out in accordance with the animal protocols approved by the Animal Care and Use Committees at the University of Tokyo and RIKEN.

**Sample preparation**. Blood was collected from BALB/c or C57BL/6 male mice at the age of 10 weeks (Japan CLEA and Charles River Japan). Frog blood was collected from female *Xenopus laevis*. Fish blood was kindly provided by M. Masuda at RIKEN CBS. Blood cells of these strains were dialyzed against distilled water in order to extract hemoglobin. Stored blood from horse and guinea pig (Sigma) were also dialyzed against distilled water. Purified rat and human hemoglobin (Sigma) were also used in our histological analysis. Small cotton balls were soaked in these samples and presented to mice 90 min before their dissection.

**Purification and characterization of active compound**. Blood from male mice was diluted with distilled water (10-fold dilution). The blood cell fraction was dialyzed with a cellulose tube (Sanko) and loaded onto TSK-GEL DEAE-5PW columns (7.5 φ × 75 mm, TOSOH). Compounds were eluted under a gradient of 0–500 mM NaCl in 20 mM Tris-HCl (pH 8.0) at 1 ml min⁻¹ and detected by Diode Array Detector L-2455 (Hitachi). Active fractions were incubated with an equivalent volume of 0.085% TFA (trifluoroacetic acid) solution and loaded onto a reverse-phase C4 HPLC column (PEGASIL-300 C4P, 4.5 φ × 250 mm, Sensyu). Samples were eluted under a gradient of 30–60% ACN (acetonitrile) in 0.085% TFA at 1 ml/min and detected by Diode Array Detector L-2455 (Hitachi). The purity of fractionated samples was confirmed by SDS-PAGE with 15% acrylamide gel, stained with Coomassie Brilliant Blue.

**Production of recombinant β-globin**. Total RNA was prepared from the liver of BALB/c mice using TRIzol reagent (Invitrogen, #15596026). β-globin cDNA was obtained by RT-PCR using the following primers:

5′- ATTCATATGGTGCACCTGACTGATGCTGAGAAGG;
5′- AATCTCGAGGTGGTACTTGTGAGCCAGGGCAGCAGC.

The amplified DNA was subcloned into the expression vector *pET-22b* (Novagen, #69744). The expression construct was transformed into *E. coli* BL21 (DE3). Constructs with single amino acid mutations (G17A and H78N) were made using specific PCR primers. Peptide expression was induced with isopropyl thiogalactoside for 4 h. Bacterial pellets were resuspended in urea buffer (5 M urea, 20 mM Tris-HCl [pH 7.5]) and sonicated. After centrifugation, the bacterial pellets were resuspended in a specific buffer and purified using the His-Bind purification kit (Novagen, #69864) in accordance with the manufacturer's protocol. After purification, recombinant protein was applied to a reverse-phase C4 column (PEGASIL-300 C4P, 4.5 φ × 250 mm, Sensyu) in order to desalt the elution buffer. The fraction with significant absorbance was collected and freeze-dried using a freeze dryer (Tokyo Rikakikai, EYELA FDU-2200). Recombinant β-globin was stored at −80 °C before use and used with a cotton ball for histological analysis, as shown in Fig. 1h.

**Purification of hemoglobin**. For hemoglobin purification, the blood of BALB/c or C57BL/6 male mice at the age of 10–12 weeks (about 500 μl in each purification) was collected and centrifuged for 10 min. A similar volume of PBS (about 250 μl) was added to the blood cell fraction. After repeating these steps twice, the same volume of distilled water was added to the blood cells. To remove residue, the blood cell solution was centrifuged again for 10 min. The solution was then diluted with distilled water (10-fold dilution). The concentration of hemoglobin was estimated to be 15 μg μl⁻¹. The hemoglobin solution was stored at −80 °C before being used for electrophysiology (Fig. 1i, j), identification of its receptor (Fig. 3a-e and Supplementary Fig. 3a), histological analysis in higher brain regions (Fig. 4 and

Supplementary Fig. 5), and behavior assays (Figs. 5, 6, 7 and Supplementary Figs. 6–8).

**Electrophysiology**. Electrovomeronasogram recording (Fig. 1i, j) was performed as described previously with minor modifications[13,39]. For the EVG recording, 10-week-old BALB/c female mice were anaesthetized and sacrificed by quick decapitation. VNO with exposed vomeronasal neuroepithelium was put in a recording chamber filled with Ringer's solution (140 mM NaCl, 5.6 mM KCl, 5 mM HEPES, 2 mM pyruvic acid sodium salt, 1.25 mM KH₂PO₄, 2 mM CaCl₂, 2 mM MgCl₂, 9.4 mM D-glucose (pH 7.4)). The field potential was recorded as previously described[9]. Spikes were analyzed using Igor Pro functions (Wave Metrics)[40]. The purified β-globin was also dialyzed against Ringer's solution before use for electrophysiology. External fluid from dialysis was used as a control.

**Histochemistry**. To prepare the sections for in situ hybridization (ISH) and immunohistochemistry, 10–12-weeks old BALB/C female (Figs. 1, 2, 3 and Supplementary Figs. 1, 2, 3) and C57BL/6 male mice (Fig. 4 and Supplementary Fig. 4, 5) were anesthetized with a lethal amount of sodium pentobarbital, sacrificed, and perfused with phosphate-buffered saline (PBS) followed by 4% paraformaldehyde (PFA) in PBS. Snouts and brain tissues were postfixed with 4% PFA in PBS overnight. To prepare VNO sections (Figs. 1a-b, 1h. 3a-e, and Supplementary Figs. 3a, 4a), snouts were decalcified in 0.5 M EDTA (pH 8.0) for 48 h at 4 °C. The tissues were then cryoprotected with 30% sucrose solution in PBS at 4 °C for 24–48 h. After collecting 14 μm coronal sections of the VNO or 30 μm sagittal sections of the AOB and 40 μm coronal sections of the brain using a Cryostat (model #CM1860, Leica), the sections were placed on MAS-coated glass slides (Matsunami).

Cryosections of the VNO and AOB were incubated with anti-c-Fos antibody (Oncogene (Ab-2), (1:1000, lot# 21584-1), Abcam (Ab-5), (1:100, lot# ab7963-1), and Calbiochem (Ab-5), (1:10,000, lot# 34095), followed by biotinylated goat anti-rabbit IgG secondary antibody (1:200, Vector Laboratories), ABC amplification (1:100, Vector Laboratories), and staining with 3,3' diaminobenzidine (Sigma). The sections shown in Fig. 1b were incubated with anti-pS6 ribosomal protein (S235/236) antibody (1:1000, Cell Signaling, #4858) and anti-Gαo antibody (1:500, MBL, #551) and Gαo signals were visualized with Alexa488-conjugated goat anti-rabbit secondary antibody (1:500, Invitrogen, A11034). The VNO sections from *Vmn2r88*-knockout mice shown in Fig. 3d-e were incubated with anti-pS6 antibody (1:200, Cell Signaling, #4858) and anti-Vmn2r88 antibody (1:300, originally made in this study) and visualized with Alexa Fluor 488-conjugated goat anti-guinea pig IgG secondary antibody (1:500, Invitrogen, A11073) and Cyanine3-conjugated goat anti-rabbit IgG secondary antibody (1:500, Invitrogen, A10522). Guinea pig antisera was raised against synthetic peptides specific to Vmn2r88: NH₂-C + IRKYKDKFRY-COOH. Antibodies were then affinity purified using affinity columns (Sulfolink) conjugated with the synthetic peptide.

Double ISH in the VNO section for receptor screening was performed as follows[10–12,14]. To synthesize the cDNA for the *V2R* and *Egr1* probes, total RNA was prepared from VNOs collected from 9 weeks old BALB/c female mice using TRIzol reagent (Invitrogen, #15596026). After RQ1 RNase-Free DNase (Promega, M6101) treatment, total cDNA was synthesized using Superscript III (Invitrogen, #18080093). cDNA for the *V2R* and *Egr1* probes was obtained by RT-PCR. The *V2R* and *Egr1* probes are both approximately 800 bp in length and the *Egr1* probes consists of 3 probes to cover nearly the full length mRNA. ISH probes were prepared by in vitro transcription with DIG RNA Labeling Mix (Roche Applied Science, #11277073910) or Fluorescein RNA Labeling mix (Roche Applied Science, #11685619910) and T7 polymerase (Promega, #P2075), T3 polymerase (Roche Applied Science, #11031163001) or SP6 polymerase (Roche Applied Science, #10810274001). *V2R* probes were labeled with DIG and *Egr1* probes were labeled with Flu unless otherwise noted. Sections of VNO underwent ISH at 60 °C or 68 °C overnight. 300 ng ml⁻¹ of *Egr1* probes and 800 ng ml⁻¹ of *V2R* probes were suspended in hybridization solution unless otherwise noted. After a series of post-hybridization washing and blocking, Flu-positive cells were visualized with anti-FITC antibody (PerkinElmer, #NEF710001EA, 1:250 in blocking buffer) followed by TSA biotin amplification reagent (PerkinElmer, #NEF749A001KT, 1:50 in 1 × plus amplification diluent) and streptavidin Alexa488 (Invitrogen, #S11223, 1:250 in blocking buffer). DIG-positive cells were visualized with anti-DIG antibody (Roche Applied Science, #11207733910, 1:250 in blocking buffer) and TSA Cy3 amplification regent (PerkinElmer, #NEL744001KT, 1:100 in 1 × plus amplification diluent). Sections were counterstained with or without 4',6-diamino-2-phenylindole dihydrochloride (DAPI, Sigma-Aldrich, #D8417) and mounted with a cover glass using Permaflour (ThermoFisher, #TA-006-FM) or Fluoromount (Diagnostic BioSystems, #K024).

In *Vmn2r88* ISH and pS6 immunostaining of VNO sections, after final washing of ISH the sections were incubated with pS6 antibody (Cell Signaling Technology, cat# 4856 S; 1:200 in blocking buffer) at 4 °C overnight, and signals were visualized with Alexa 488-conjugated goat anti-rabbit secondary antibody (Invitrogen, A11034).

In situ hybridization of *c-Fos* brain mapping was performed as follows[14]. A DIG-labeled probe for *c-Fos* was previously characterized[12,14]. The *c-Fos* probe was prepared by in vitro transcription with a DIG-RNA labeling mix (#11277073910) and T3 RNA polymerase (#11031163001) in accordance with the manufacturer's

instructions (Roche Applied Science). Target brain regions underwent ISH at 60 °C overnight. After a series of post-hybridization washing and blocking, DIG-positive cells were visualized with anti-DIG antibody (Roche Applied Science, #11207733910, 1:250 in blocking buffer) and TSA-plus Cyanine 3 (PerkinElmer, #NEL744001KT, 1:100 in 1 × plus amplification diluent). Sections were counterstained with DAPI (Sigma-Aldrich, #D8417) to visualize the nuclei and then mounted with cover glass using Fluoromount (Diagnostic BioSystems, #K024). This method was also applied for post-hoc c-Fos ISH staining after weak light stimulation onto SF1-positive neurons of lactating female mice (Supplementary Fig. 11a-b). For double ISH staining with c-Fos and SF1 probes (Supplementary Fig. 9a-b), SF1 probe was prepared as previously mentioned[12] and its procedure was basically the same as double ISH in the VNO sections.

Imaging of the sections was performed with an Olympus BX53 microscope (10 × or 20 × objective) equipped with an ORCA-R2 cooled CCD camera (Hamamatsu Photonics). Images were processed using Adobe Photoshop CS2 or CS6 (Adobe Systems)[41]. For cell counting, the number of sections shown in figure legends of Fig. 4 and Supplementary Fig. 5 was used for each brain region to cover entire populations from anterior to posterior.

**Generation of mutant mice by CRISPR-mediated genome editing**. To generate a null mutant of Vmn2r88, we designed two guide RNAs (gRNAs) for each gene that were able to introduce double-strand DNA breaks flanking exon 3-6 of the V2R gene in which the essential transmembrane domain is encoded. CRISPR-mediated genome editing was performed as described previously with minor modifications[14,15].

Cas9 mRNA was prepared as follows[42]. pMLM3613 (Addgene, #42251) was digested with PmeI and purified with ethanol precipitation. In vitro transcription was performed using the mMESSAGE mMACHINE T7 ULTRA Transcription Kit (ThermoFisher Scientific, #AM1345) in accordance with the manufacturer's instructions. The amount and purity of synthesized mRNA were tested using electrophoresis with a 1% agarose gel. To design gRNAs to target Vmn2r88, we first searched for 20 bp target sequences upstream of the protospacer adjacent motif (PAM) using CRISPR-direct (http://crispr.dbcls.jp/). We then selected a target sequence with > 50% CG content that was completely unique in the mouse genome (confirmed by GGGenome at http://gggenome.dbcls.jp/ja/mm10/2/). The selected sequence was then introduced into the BsaI-digested pDR274 construct (Addgene, #42250) using the following oligo-DNAs:

*Vmn2r88* upstream
5′-TAGGCGTAGATGTACACTGCAAAC;
5′-AAACGTTTGCAGTGTACATCTACG.
*Vmn2r88* downstream
5′-TAGGAGAACCAGGAATCTCAACTG;
5′-AAACCAGTTGAGATTCCTGGTTCT.

After validating the sequence, pDR274 with the target DNA sequence was digested with DraI, and in vitro transcription of gRNA was performed using the MEGA shortscript T7 Transcription Kit (ThermoFisher Scientific, #AM1354) in accordance with the manufacturer's instructions. The synthesized gRNA was purified using the MEGAclear Transcription Clean-Up Kit (ThermoFisher Scientific, #AM1908). The amount and purity of the synthesized gRNA were tested using electrophoresis with a 1% agarose gel. A mixture of 20 ng μl$^{-1}$ of two gRNAs and 50 ng μl$^{-1}$ of Cas9 mRNA was injected into C57BL/6 J fertilized eggs in order to generate the knockout mice. The genotypes of the mutant mice were determined by two kinds of PCR methods using the following oligo-DNAs.

5′-GCATTCTTCAATGCCACTGGTAAG;
5′-AATCTGCGGTGTGCAAAAGT;
5′-GCAGCCACTCCATGAAAGCA.
Mutant allele = 450 bp
5′-CGTAGATGTACACTGCAAACAGG;
5′-CTTCTGCATGCACTCATGTACC;
Wild type allele = 3000 bp

**Behavior assays**

*Digging assay*. The digging behavior assay (Fig. 5) was performed in the test mice's home cage. Their home cages were moved to a recording space and their food and water were temporarily removed 1 h before the recording. In the assays shown in Figs. 5b-c, 5e-f, their pups were also removed temporarily. The mice were exposed to a cotton ball, with or without hemoglobin (300 μg), ESP1 (20 μg), or diluted 2MT (from 10-fold to 50,000-fold in mineral oil). The number of digging behaviors, duration of total digging time, and other parameters were calculated for a 20 min recording.

*Pup retrieval assay*. For the pup retrieval assay (Supplementary Fig. 7), C57BL/6 J mothers with their pups (postnatal day4-6) were used. All of the pups were removed from their home cages 30 min before behavior recording and a stimulant (30 μL of fresh blood, hemoglobin (300 μg) or distilled water) was painted on their backs. Then, three pups were placed in each corner of the cage, excepting the one corner which was nearest to the nest. Their behaviors were recorded for 30 min to observe not only pup retrieval but other behaviors after their retrieval.

*Open field assay*. The open field assay (Fig. 6a-c) was performed for 10 min in a 40 cm × 40 cm × 40 cm square open arena under normal lighting. C57BL/6 J lactating mothers with pups were presented with a cotton swab, with or without hemoglobin (300 μg), ESP1 (20 μg), or 2MT (Sigma-Aldrich, 1:10 or 1:10,000 dilution in mineral oil) for 5 min and moved into the test arena just before testing. The movement of the mice was videotaped and scored for the following parameters: rearing time, total distance, center time and moving speed. The parameters without rearing time were analyzed by ImageJ software (version 2.1.0)[43].

*Two-chamber test*. The two-chamber test (Supplementary Fig. 8a-c) was conducted between 2 and 10 h after the start of the light period. Initially, animals were transferred into a 25 cm × 50 cm × 25 cm behavior chamber with two rooms in dim light conditions. Animals (lactating female mice) were kept in the cage for 5 min for habituation. After habituation, a piece of filter paper (5 cm × 5 cm) soaked with either 100 μL of water, water containing 300 μg of hemoglobin, or 2MT diluted in mineral oil (1:10,000), was placed on one side of the chamber. Animal behavior was recorded for 10 min by a USB camera (logicool). Each animal went through 3 trials, with different stimuli, within 4 days. The trajectory of the animal, locomotion, and the total time spent in each room, was quantified using a custom written Python program.

**Pharmacogenetics**. For chemogenetic inhibition of SF1 + neurons in the VMHd (Fig. 7a-c), we prepared SF1-Cre female mice and conducted stereotactic surgery as described in the **Optogenetics**. For control group (Supplementary Fig. 10), we purchased parous wild type C57BL/6 female mice from Japan CLEA and performed stereotactic surgery after one additional weaning experience. We injected AAV8-hSyn-DIO-hM4D(Gi)-mCherry (Addgene, lot# v62036, ~ 2.4 ×10$^{13}$ gp mL$^{-1}$) (300–350 nL, 40 nL min$^{-1}$) into the bilateral VMHd. Two days after surgery, female mice were paired with stud C57BL/6 J male mice to engage in mating. Once females got pregnant, we removed paired males. After the third parturition, lactating mothers raising pups (postnatal day 3–9) were subjected to the behavioral assay. One day before the behavioral test, we changed beddings and injected 0.2 mL of saline intraperitoneally to acclimate the intraperitoneal injection. We performed a behavioral assay as described in the *Digging assay* with some modifications. Sixty minutes before the beginning of the behavioral testing, 0.2 mL of 0.1 mg mL$^{-1}$ clozapine-N-oxide dissolved in saline (CNO, Sigma-Aldrich, cat#C0382) or saline was administered intraperitoneally to the subject female mice, and pups, food, and water were removed. The mice were exposed to a cotton ball with hemoglobin (300 μg). Each animal underwent a single behavioral assay because very few mice bite a cotton ball after second assays. The duration of total digging time and rearing were quantified for a 20 min recording. Each animal was checked post-hoc to determine if the viral injection was correctly administered.

**Optogenetics**. For optogenetic neural activation of SF1 + neurons in the VMHd (Fig. 7d-f and Supplementary Fig. 9a-b, 11a-g), we prepared SF1-Cre female mice which experienced 2 times of parturition. These SF1-Cre female mice were anesthetized with 65 mg kg$^{-1}$ ketamine (Daiichi-Sankyo) and 13 mg kg$^{-1}$ xylazine (Sigma-Aldrich) via intraperitoneal injection and head-fixed to stereotactic equipment (Narishige). Then, we injected AAV5-EF1a-DIO-hChR2(H134R)-eYFP (UNC vector core, lot# AC4313U, ~ 5.5 ×10$^{12}$ gp mL$^{-1}$) or AAV8-CAG-DIO-GFP (UNC vector core, lot# AV4910b, ~ 6.2 ×10$^{12}$ gp mL$^{-1}$) (300–350 nL, 40 nL min$^{-1}$), as a control virus, to the unilateral VMHd (Posterior, 1.0 mm; Lateral, 0.25 mm; Ventral, 5.25 mm, from the Bregma), using a UMP3 pump regulated by a Micro-4 device (World Precision Instruments). Soon after the virus injection, optical fibers (200 μm core, 0.39-NA) (Thorlabs, cat#FT200UMT) were implanted 200 μm above the VMHd unilaterally. Two days after surgery, female mice were paired with stud C57BL/6 J male mice to engage in mating. Once females got pregnant, we removed paired males. After the third parturition, lactating mothers raising pups (postnatal day3-9) were subjected to the behavioral assay. Behavioral experiments were performed during the light period. To observe a light stimulation effect, animals were placed into a 30 cm × 30 cm × 30 cm chamber with 5 cm thick bedding and connected to a 473 nm laser (Changchun New Industries) through a rotary joint patch cord (Thorlabs, RJPFL2) and cannula. We recorded behaviors during two successive light conditions, 5 min of light off phase and 5 min of light on phase (0 mW, 0.01 mW, 0.03 mW, or 1 mW). These behavioral tests were conducted on 2 separate days and the interval of the behavioral test was more than 2 h. Light delivery was controlled using an Arduino microcontroller (ARDUINO ZERO #ABX00003) and simple custom-made code[44]. Animal behavior was recorded by a video camera (logicool) from a horizontal and vertical view, and digging behavior was analyzed. Each animal was checked post-hoc to determine if the viral injection and fiber positioning were correctly administered.

**Quantification and statistical analysis**. Data are presented as mean ± S.E.M. unless otherwise noted. The statistical details of each experiment, including the statistical tests used, the exact value of n, and what n represents, are detailed in each figure legend. The two-sided Wilcoxon rank-sum test was used for tests shown in Fig. 4b-c, e-f, h, and the two-sided Wilcoxon rank-sum test with Dunnett correction in Fig. 6b. The two-sided Steel-Dwass test was used for tests shown in Figs. 2e, 5b-c, e-f, 6c, and Supplementary Fig. 7c. Unpaired two-sided Student's t-test was used for tests in Supplementary Figs. 6b, 10b, and 11b. One-way ANOVA

with repeated measures and Bonferroni's correction was used for the test shown in Supplementary Fig. 8c. The two-sided Wilcoxon signed-rank test was used for the test shown in Fig. 7c, f and Supplementary Fig. 11d. Significance was noted as **$p < 0.01$, and *$p < 0.05$. R version 3.5.0 and python 3 were used for all non-parametric statistical analyses in this study[45,46].

**Reporting summary**. Further information on research design is available in the Nature Research Reporting Summary linked to this article.

## Data availability
All relevant data is available from the authors with reasonable request. The structure of human hemoglobin shown in Fig. 2f is from RCSB Protein Data Bank (https://www.rcsb.org/structure/1a3n). Source data are provided with this paper.

## Code availability
All original codes are available from the authors with reasonable request.

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

## Acknowledgements
We thank RIKEN CBS Research Resource Division for help in generating knockout mice, and members of the Touhara lab for their helps. We thank T. Kikusui (Azabu Univ.) for valuable discussion. T.O., T.A., and K.K.I. are supported by research fellowship for young scientist from JSPS. This work was supported by Grants-in-Aid for Scientific Research (S) (Grant Numbers 24227003 and 18H05267) and Young Scientists (S) (Grant Number 19677002) from JSPS Japan and ERATO Touhara Chemosensory Signal Project from JST, Japan (Grant Number JPMJER1202) to K.T., and JSPS Kakenhi (Grant Number 16K20963 and 17H05552) to K.Miyamichi.

## Author contributions
T.O., T.A., K.K.I., H.K., K.Miyamichi, and K.T. designed the study. T.O. performed histochemical analyses and behavior assays. T.A. identified hemoglobin and interaction

site of hemoglobin with its receptor. H.M. performed behavior assay and histochemical analyses with a help from K.Miyamichi. T.I. performed chemogenetic inhibition assay with a help from K.Murata. K.K.I., T.I., and K.Murata performed optogenetics. R.E. generated mutant mice with a help from Y.Y., K.Murata measured corticosterone. T.I., K.Murata, and K.S. performed some parts of histochemical analyses and behavior assays. S.H.-Y., and H.K. provided preliminary suggestive data of this study. T.O. and K.T. wrote the paper with substantial contributions from other authors.

## Competing interests

The authors declare no competing interests.

## Additional information

**Peer review information** *Nature Communications* thanks Cory Root and the other anonymous reviewer(s) for their contribution to the peer review this work. Peer reviewer reports are available.

