## [Peer Review File · Nature Communications]

Reviewers' Comments:

Reviewer #1:

Remarks to the Author:

In the manuscript from Osakada et al., the authors define molecular mechanisms for mouse detection of blood. They nicely, identify a single receptor expressed in the vomeronasal organ (VNO) that detects a particular amino acid residue of hemoglobin. Further they show that detection of hemoglobin triggers apparent exploratory behaviors specifically in lactating females. Blood as a sensory cue is somewhat novel and it is not entirely clear what survival advantage is afforded by this sensory pathway in mice, but it is interesting that it is unique to lactating females. This part of the manuscript is solid and very well done. They move on to look at downstream brain areas involved in the behavioral response to hemoglobin, by examining hemoglobin-evoked c-fos activity, and they find activity in typical parts of the VNO pathway. They then, activate neurons in the ventromedial hypothalamus (VMHd) and find that weak activation produces behaviors similar to those elicited by hemoglobin. The implication of the neural circuit involved is a quite weak. Nonetheless, the finding of the receptor and ligand that play a role in a novel behavior, is well done and interesting. I offer the following, mostly minor concerns and suggestions.

1. In Figure 2c. Why do the authors not show the amino acid sequence from frog and zebra fish to be consistent with Figure 2b? Presumably, they also lack the G17 amino acid. Although this is really minor, it would be nice to see and it would be more continuous with the previous graph.
2. Figure 3b&c: It is unclear what the letters along the X-axis indicate. Presumably these are different clades in (b) and different receptors in (c), but this should be clarified in the legend or text.
3. Figure 3d: It is difficult to see the pS6 immunoreactivity, and nearly impossible in the merged image. The authors should adjust the brightness/contrast to make it more clear.
4. Figure 5e&f: The "indicated concentrations" are not clear. Do these numbers represent dilutions?
5. In Figure 6, the authors use an open field assay to ask if hemoglobin triggers anxiety like behaviors. They do not observe any clear signs of anxiety suggesting that pre-exposure to hemoglobin does not produce anxiety. However, without a positive control it is unclear if the assay is suitable to detect pre-exposure evoked anxiety. The authors should show, for example, that pre-exposure to 2MT causes some measure of anxiety in this assay. In addition, it would be useful to look at hemoglobin exposure during the assay. What if hemoglobin causes a transient state of anxiety that is reset by introduction to the assay?
6. The weakest part of the paper is Figure 7, which could be eliminated and included in a future paper about the neural circuit involved in this behavior. The authors activate SF1-positive neurons in VMHd and show that weak activation evokes digging responses similar to that by hemoglobin. First, it is unclear why they focus on SF1 positive neurons other than SF1 is expressed in VMHd. Are these SF1 neurons activated by hemoglobin? Second, this activation experiment does not tell us anything about whether these neurons are involved in the hemoglobin-evoked behavior. If the authors wanted to make this claim, they should show loss of function data. Third, the optical stimulation is unusually low intensity (0.01-0.03 mW, whereas most people use 1-5 mW). Does this even activate the neurons? Lastly, the model of the neural pathway (Figure 7f) is not well supported and only loosely defined by c-fos expression and previous anatomical work. Much more work needs to be done to clearly test if this model is correct.

Cory Root

Reviewer #2:

Remarks to the Author:

This elegant study identifies a specific receptor in the vomeronasal organ (VNO) of mice, Vmn2r88,

that is sensitive to blood. The authors further demonstrate that the amino acid Gly17 in the hemoglobin protein is crucial for the interaction between blood and the receptor Vmn2r88. Furthermore, the authors show that hemoglobin acts as a chemosensory signal to the brain that has an important behavioral function. They do so by showing that hemoglobin exposure results in "exploratory" behaviors in lactating mothers.

Even though the G protein-coupled receptor Vmn2r88 was recently identified as important for hemoglobin sensing in mice (Isogai et al., Cell 2018), the authors make additional important contributions that make the first part of the paper particularly impressive. Specifically, the authors show that Gly17 on hemoglobin is a crucial site for receptor interactions. This finding might also provide an evolutionary perspective on blood sensing and is crucial for understanding the interaction between hemoglobin and the Vmn2r88 receptor. Another central addition to the Isogai et al., paper is the second part of this study, which identifies a behavioral output following the olfactory sensing of hemoglobin, and identifies potential downstream neuronal circuits that are important for executing the behavioral outputs. This is an important new discovery as in Isogai et al., the authors could not identify a major behavioral affect when mutating the Vmn2r88 receptor. Taking these new discoveries into consideration, this paper can to be suitable for publication in Nature Communications if some further work would be added to strengthen the second part of this study (see below).

Major concerns:

1. Mothers and non-parental mice. While most of the major new findings in the second part of the paper are performed in lactating females, the optogenetics and experiments of Figs. 1-3 are performed in naïve males. At the very least, the optogenetic experiments should be performed in lactating females as well. Furthermore, it would be helpful to know if the Hb fos responses are comparable in the VNO \ AOB of mothers vs. naïve females. Such data could add mechanistic insights to the changes in responses to hemoglobin observed between naïve females and mothers in downstream brain regions. Other work on the accessory olfactory system seem to support this line of inquiry (Dey et al., Cell 2015, Xu et al., Neuron 2016).

2. Behavioral responses to hemoglobin. In order to identify how hemoglobin affects behaviors through neuronal activity, the authors show: 1. That exposure to hemoglobin results in elevation of cfos expression in VMHdm of mothers. 2. That hemoglobin exposure results in digging and rearing behaviors. 3. That artificial activation of SF1+ neurons in VMHdm can result in behaviors such as digging and rearing in naïve males. However, this does not clarify the question of whether or not hemoglobin actually elicits digging\rearing behaviors through SF1+ neurons in VMHdm in mothers. In order to connect these observations, the authors should use a loss of function approach. Ideally, the authors should reversibly silence SF1+ neuronal activity in VMHdm and test whether the behavioral responses to hemoglobin are affected. This can potentially show that SF1+ neurons in VMHdm are necessary for executing rearing\digging behaviors upon sensing hemoglobin.

3. "Risk assessment" vs. "exploratory" behavior. The authors use the term "exploratory behaviors" when they describe the behavioral outputs observed in response to hemoglobin (in mothers), based on negative state (e.g., cort levels). However, it can be argued that the authors could not detect anxiety markers because the anxiety levels are too low to produce detectable increases. Furthermore, the behavioral data that are presented in the paper may be more accurately described as risk assessment, a mild form of defensive behavior or what Fanselow and Zhuravka described as an "early defensive mode" (Fanselow and Zhuravka, Behavioral Processes 2019). Importantly, it has been demonstrated earlier that the VMHdm and specifically SF1+ neurons in VMHdm are important in processing defensive behaviors including rearing (Kunwar et al., eLIFE 2015, Wang et al., Neuron 2015). In addition, digging can be considered as a form of a risk assessment (for example Gozzi et al., Neuron 2010). The authors should therefore consider switching\adding the term "risk assessment" in order to be more consistent with the literature (Gross and Canteras, NRN 2012, Ribeiro-Barbosa et al., Neuroscience & Biobehavioral Reviews 2005). Interestingly, the authors show using an innate fear inducing volatile odor (2MT, fig.5f), that mice transit from an immobility\freezing behavior elicited by high doses of 2MT (freezing due to immediate danger) to digging behavior ("reduce the likelihood of encountering danger" Fanselow and Zhuravka, Behavioral Processes 2019). Since both low doses of 2MT and hemoglobin elicit similar risk assessment\exploratory behaviors, it would be informative to examine whether a lower dose 2-MT activates the same SF1+ VMHdm neurons as Hb in mothers (e.g., by fos-catFISH or single-cell imaging). This is especially important since TMT, a compound related to 2-MT, has previously been shown not to activate VMHdm (Perez-Gomez et al., 2015), in contrast to other

predator odors.

Minor concerns:

- It has been demonstrated that outputs from VMHdm to AHN are important for avoidance behaviors, and that VMHdm to dPAG is important for immobility (Wang et al., Neuron 2015). The authors could address these findings made by previous works through examining cFOS expression in the AHN of mothers (and if needed, change figure 7f accordingly).
- In several places, the authors measure cfos activity "Per region". This is not clear. Did the authors collect the whole region? Did the authors randomly apply a single brain slice from each region? Since many of the observations described here rely on this analysis, this should be explained in more detail and include exact coordinates.
- Why are there responses in the VNO of naïve males and females if they have no behavioral relevance? The authors should clarify this in the text.
- What is the role for the main olfactory bulb in hemoglobin sensing in mothers? This is important especially since it has been shown that there is functional plasticity to behaviorally relevant odors in the main olfactory bulb of mothers (Vinograd et al., Cell Reports 2017).
- Figure 6 b and 6 c. What is the difference between the control groups in figure 6b (rearing time) and figure 6c? Similarly, what is the difference between the hb condition in figure 6b and hb +/- condition in figure 6c? If these are different animals for the same conditions the authors should group them together to increase the statistical power.
- Figure 1, individual mice. While in all figures the variance can be estimated by explicitly showing the data for individual subjects on top of the bar graphs, it will be informative to add this in figure 1 as well (and not just error bars).
- In supplemental figure 7, mice spent similar time in a 2-chamber assay where hemoglobin was compared to the control. However, several concerns arise from this interpretation made by the authors. Most importantly, it seems that the mice tested in the assay were male mice and not mothers. This is a surprising choice for this assay, considering the fact that males did not show a significant behavioral difference in other assays, and other works imply that hemoglobin might be attractive to males (Isogai et al., Cell 2018).

In summary this is a strong submission and our comments should be taken as constructive remarks to improve the paper and eliminate confusion and ambiguity for the readers. We congratulate the authors on their beautiful and impressive work.

Reviewer #3:

Remarks to the Author:

In this paper, Osakada and colleagues – stimulated by initial observations that blood activates the VNO – identify the specific blood ligand responsible for this activation (beta-globin) and its corresponding VNO receptor (Vmn2r88). Using Vmn2r88-knockout mice, they demonstrate that smelling blood causes exploratory behavior (rearing and digging) in a receptor-dependent manner, and that this response is elicited in lactating mothers, but not virgin females or males. The behavioral response is attributed to the specific activation of dorsal PAG, MeA and dorsal VMH (dVMH) in lactating mothers, and was recapitulated by optogenetic activation of SF1 neurons in dVMH. On the basis of these data, the authors claim that the smell of blood activates a specific behavioral response that is dependent on an AOB-MeApv-VMHd-PAGd pathway and unique to lactating mothers.

Overall, the experiments identifying beta-globin and Vmn2r88 (Figs 1 and 3) in this paper are elegant and convincing, and consistent with a recent paper (Isogai, 2018) that also found beta-globin and Vmn2r88 to mediate the sensing of blood VNO. The subsequent neural and behavioral analysis, however, could be improved to rule out alternative explanations for the authors claims. In particular, I am concerned that the data may reflect a general response to VNO activation that could include a much larger category of ligands than just beta-globin. For example, the authors show that 2MT also elicits digging behavior when presented at very low concentration. In addition to this overall feedback, I have several concerns and questions that will clarify this manuscript:

1. To clarify the specific effects of hemoglobin, it would be helpful to see the same behavioral experiments repeated for other ligands, including some that would likely covary with hemoglobin (e.g. other infant-associated ligands), and others that are independent.
2. Fig 7 shows the result of optogenetic stimulation of SF1 neurons in dVMH. SF1 neurons are allegedly involved in fear responses and comprise the majority of dVMH cells. Interpreting this result requires knowing whether blood activates these SF1 neurons.
3. Figure 2d,e is used to show that the H78N mutation in beta-globin does not inhibit VNO activation. However, data are only shown for two mutant proteins and a no-odor control, but not for the WT protein. Thus it cannot be determined from the figure whether H78N leads to a reduction of VNO activation compared to the WT protein.
4. Is the ability of VNO neurons to detect hemoglobin state dependent (like the fos activity reported in the brain)?
5. If blood is placed on a pup, how does the mother behaviorally react?

We greatly thank all reviewers for the enthusiasm about our study, and for their constructive comments. We have performed additional experiments and modifications in response to the comments, which have resulted in significant improvement of the paper. Below we first summarize the additional experiments in this revision, and then provide point-by-point responses to the reviewer's comments.

Additional experiments performed:

- **Adding new conditions to open field, digging behavior, and pup retrieval assays with lactating mothers.** (Revise figure 2, 10, and 12)
- ***In situ* hybridization with brain sections including the VMHd to see whether hemoglobin can activate SF1-expressing neurons in lactating females.** (Revise figure 3)
- **Pharmacological loss-of-function assays in SF1-expressing neurons in the VMHd.** (Revise figure 4)
- **Post-hoc *c-Fos in situ* hybridization after weak light stimulation on SF1-positive neurons in the VMHd.** (Revise figure 5)
- **Updated optogenetic experiments of SF1-positive cells in the VMHd with lactating female mice.** (Revise figure 6)
- **Staining of the VNO and AOB sections to see responses towards hemoglobin in peripheral neurons in both lactating and virgin females.** (Revise figure 7)
- ***In situ* hybridization with brain sections including the VMHd to see whether a low concentration of 2MT can activate neurons in the VMHd.** (Revise figure 8)

Point-by-point responses:

Reviewer #1 (Remarks to the Author):

In the manuscript from Osakada et al., the authors define molecular mechanisms for mouse detection of blood. They nicely, identify a single receptor expressed in the vomeronasal organ (VNO) that detects a particular amino acid residue of hemoglobin. Further they show that detection of hemoglobin triggers apparent exploratory behaviors specifically in lactating females. Blood as a sensory cue is somewhat novel and it is not entirely clear what survival advantage is afforded by this sensory pathway in mice, but it is interesting that it is unique to lactating females. This part of the manuscript is solid and very well done. They move on to look at downstream brain areas involved in the behavioral response to hemoglobin, by examining hemoglobin-evoked *c-fos* activity, and they find activity in typical parts of the VNO pathway. They then, activate neurons in the ventromedial hypothalamus (VMHd) and find that weak activation produces behaviors similar to those elicited by hemoglobin. The implication of the neural circuit involved is a quite weak. Nonetheless, the finding of the receptor and ligand that play a role in a novel behavior, is well done and interesting. I offer the following, mostly minor concerns and suggestions.

We thank the reviewer for careful read of the manuscript and important suggestions.

1. In Figure 2c. Why do the authors not show the amino acid sequence from frog and zebra fish to be consistent with Figure 2b? Presumably, they also lack the G17 amino acid. Although this is really minor, it would be nice to see and it would be more continuous with the previous graph.

We thank the reviewer for this suggestion. We added the sequences of frog and zebrafish β -globin (**new Figure 2c**). The sequences show that there is no G17 in not only β -globin of BALB/c minor and horse but that of frog and zebrafish, all of which did not show β -globin-dependent c-Fos enhancement (**Figure 2b**).

2. Figure 3b&c: It is unclear what the letters along the X-axis indicate. Presumably these are different clades in (b) and different receptors in (c), but this should be clarified in the legend or text.

We are sorry for the unclear description. In V2Rf clade, there are 21 V2R receptors, so we performed double *in situ* hybridization three times with different probes (1st: probes for each V2R clade [V2Ra, b, e, f, i, j, k, l, n, o, p, q, s, u (Tsunoda et al.¹, Osakada et al.²), 2nd: probes to narrow down in V2Rf clade [using probes named V2Rf1 to V2Rf5], 3rd: probes to distinguish each gene in V2Rf5 clade [Vmn2r88, 89, and 122]) to find the candidate receptor gene. We modified the description in the figure legends of **Figure 3b and 3c**.

3. Figure 3d: It is difficult to see the pS6 immunoreactivity, and nearly impossible in the merged image. The authors should adjust the brightness/contrast to make it more clear.

Thank you so much for this suggestion. We have changed the brightness and contrast of images (**Revise figure 1 below, Figure 3d** in the manuscript).

Revise figure 1

Vmn2r88 and pS6 immunostaining of VNO sections from *Vmn2r88*^{+/+} or *Vmn2r88*^{-/-} male mice exposed to hemoglobin (Hb) or distilled water.

Open arrowheads show pS6-positive cells; closed arrowheads show cells double-labeled for pS6 and Vmn2r88. Scale bar, 50 μ m. In the mutant mice, corresponding pS6 and Vmn2r88 expression completely disappeared. Scale bar, 50 μ m.

4. Figure 5e&f: The “indicated concentrations” are not clear. Do these numbers represent dilutions?

We sincerely apologize that our previous description was not precise. We used 2MT (2-Methyl-2-thiazoline, Sigma-Aldrich, M83406), diluted in mineral oil from 100-fold to 50,000-fold. We have changed the description of the *x*-axis and figure legends in **new Figure 5e and 5f**.

5. In Figure 6, the authors use an open field assay to ask if hemoglobin triggers anxiety like behaviors. They do not observe any clear signs of anxiety suggesting that pre-exposure to hemoglobin does not produce anxiety. However, without a positive control it is unclear if the assay is suitable to detect pre-exposure evoked anxiety. The authors should show, for example, that pre-exposure to 2MT causes some measure of anxiety in this assay. In addition, it would be useful to look at hemoglobin exposure during the assay. What if hemoglobin causes a transient state of anxiety that is reset by introduction to the assay?

We thank the reviewer for these important comments. We performed open field assays with pre-exposure of 2MT but there was no decrease in the total center time (**Revise figure 2 below, new Figure 6** in the manuscript), suggesting that as this reviewer pointed out, we cannot mention anxiety in this open field assay. On the other hand, rearing enhancement was observed specifically upon stimulation with hemoglobin that was diminished in the hemoglobin receptor-deficient mice, suggesting that a type of exploratory behavior was induced. It should be noted that the rearing enhancement was observed when the assay was performed without bedding, whereas digging but not rearing behavior was seen in the presence of bedding (**Figure 5 and 7 and Supplementary figure 6 and 7**), indicating that the behavioral output appears to be different depending on the assay environment.

Revise figure 2

Hemoglobin but not ESP1 and 2MT enhances rearing, a type of exploratory behavior, in lactating mothers.

a Schematic illustration of the timeline of the open field assay with cotton pre-exposure. Cotton exposure was performed in their home cage (with their pups). **b** Quantification of total distance, total center time, moving speed, and rearing time duration of lactating mothers, pre-stimulated with control buffer-, ESP1-, 2MT- (low 2MT: 10000-fold dilution, high 2MT: 10-fold dilution) or Hb-cotton swabs in the open field assay for 10 minutes. $n = 8$ for control, $n = 7$ for Hb, and $n = 5$ for ESP1, low 2MT, and high 2MT. Error bars, S.E.M. $*p < 0.05$ by Steel-test. **c** Quantification of rearing time duration of *Vmn2r88* mutant lactating mothers, pre-stimulated with control buffer- or Hb-cotton swabs. $n = 6-10$. Error bars, S.E.M. $*p < 0.05$ by the Steel-Dwass test.

6. The weakest part of the paper is Figure 7, which could be eliminated and included in a future paper about the neural circuit involved in this behavior. The authors activate SF1-positive neurons in VMHd and show that weak activation evokes digging responses similar to that by hemoglobin. First, it is unclear why they focus on SF1 positive neurons other than SF1 is expressed in VMHd. Are these SF1 neurons activated by hemoglobin? Second, this activation experiment does not tell us anything about whether these neurons are involved in the hemoglobin-evoked behavior. If the authors wanted to make this claim, they should show

loss of function data. Third, the optical stimulation is unusually low intensity (0.01-0.03 mW, whereas most people use 1-5 mW). Does this even activate the neurons? Lastly, the model of the neural pathway (Figure 7f) is not well supported and only loosely defined by c-fos expression and previous anatomical work. Much more work needs to be done to clearly test if this model is correct.

We agree with the comments. We performed some additional assays to support the conclusions. First of all, we analyzed the distribution of hemoglobin-dependent *c-Fos* positive cells in the VMHd by dual-color *in situ* hybridization with both *c-Fos* and *SF1* cRNA probes. Hemoglobin activated *SF1*-positive cells in the VMHd (**Revise figure 3 below**, added into **Supplementary figure 9** in the revised manuscript), and the number of *SF1* and *c-Fos* double-positive cells was larger than that of *c-Fos* and *SF1*-negative cells (the number of *SF1*-negative and Hb-dependent *c-Fos*+ cells; 37.8 ± 7.8 , the number of *SF1*-positive and Hb-dependent *c-Fos*+ cells; 60.3 ± 14.6 , both shown in black in **Revise figure 3b below**). These results suggest that *SF1*, whose expression is mostly limited in the VMHd (McClellan et al.³) appears to be a good molecular marker to manipulate the cells in the VMHd as shown in most of the previous studies focusing on the VMHd (Kunwar et al., Wang et al., Zhang et al.⁴⁻⁶).

Next, we performed the loss-of-function experiment using virus encoding DREADD-Gi to silence neural activities of the *SF1*-positive cells in the VMHd of lactating *SF1-Cre* female mice (**Revise figure 4 below**, added into **new Figure 7** in the revised manuscript). Virus injection into *SF1*-positive cells in the VMHd was performed before breeding. After waiting for sufficient viral infection and their childbirth, CNO or saline injection into the lactating female mice was performed 60 minutes before hemoglobin exposure and at the same time, their pups were removed from the cage. Behavioral recordings just after hemoglobin stimulation showed that neural silencing elicited a significant decrease in the digging behavior duration (**Revise figure 4c below**). These results suggest that there is the necessity of *SF1* expressing neurons in the VMHd for hemoglobin-dependent digging enhancement.

For the gain-of-function assay about *SF1*-positive neurons in the VMHd, we did post-hoc *c-Fos in situ* hybridization after weak light stimulation onto the cells (**Revise figure 5 below**, added into **new Supplementary figure 10** in the revised manuscript). Comparing to the control group with lactating *SF1-Cre* animals injected GFP-encoded virus into the VMHd, lactating females with ChR2-encoded virus injection showed a significant increase in the number of *c-Fos* expressed-cells in the VMHd after weak light stimulation (10 μ W). These results show that weak light stimulation indeed activated *SF1*+ cells, resulted in digging enhancement (**Revise figure 6 below**, **new Figure 7** in the revised manuscript).

Lastly, our model in **Figure 7f** appears to be supported by these new results, but the circuit is still loosely defined as the reviewer mentioned. Thus, we decided to move the model to **Supplementary figure 10h** to advocate potential neural circuits responsible for hemoglobin information and its outputs in lactating females.

Revise figure 3

Hemoglobin activates SF1 expressing neurons, a specific molecule marker for the sub-region, in the VMHd.

a Images from dual-color ISH staining of sections including the VMH from hemoglobin (Hb) or distilled water (control) -stimulated C57BL/6 mother mice labeled with the *SF1* cRNA probe (green) and *c-Fos* cRNA probe (magenta). Scale bar, 100 μ m. **b** Quantification of *c-Fos*-positive neurons overlapping with or without *SF1* in the VMHd. The number of sections counted to determine the number of *c-Fos*-positive neurons in each brain area was 6. Error bars, S.E.M. $n = 4$. * $p < 0.05$ by unpaired two-sided Student's t-test.

Revise figure 4

SF1-positive neurons in the VMHd are necessary for hemoglobin dependent digging behavior.

a Schematic illustration of the animal setup and timeline for pharmacological inhibition of *SF1* expressing neurons in the VMHd. AAV-DIO-hM4Di-mCherry is injected into SF1-positive cells in the VMHd. Image adapted from The mouse brain in stereotaxic coordinates (Academic Press, 2007)⁸. **b** A representative coronal section showing DREADD-Gi expression (mCherry positive cells shown in red) in the VMHd. Scale bar, 500 μ m. **c** Quantification of the digging and rearing total time duration (sec) of Hb stimulated *SF1-Cre* lactating

mothers with pre-saline i.p. injection and CNO i.p. injection. $n = 4$. Error bars, S.E.M. $*p < 0.05$ by the Wilcoxon signed-rank test.

Revise figure 5

Weak light stimulation evokes *c-Fos* activation in SF1-positive neurons in the VMHd.

a Representative images of *c-Fos* ISH staining of brain sections from blue laser-stimulated (0.01 mW, 5 min) *SF1*-Cre mother mice injected with *AAV-DIO-GFP* (SF1-EYFP, $n = 4$) or *AAV-DIO-ChR2* (SF1-ChR2, $n = 6$) in the VMHd. Scale bar, 100 μ m. **b** Quantification of *c-Fos*-positive neurons in the VMHd from fiber trace containing sections. Error bars, S.E.M. $**p < 0.01$ by unpaired two-sided Student's *t*-test.

Reviewer #2 (Remarks to the Author):

This elegant study identifies a specific receptor in the vomeronasal organ (VNO) of mice, *Vmn2r88*, that is sensitive to blood. The authors further demonstrate that the amino acid Gly17 in the hemoglobin protein is crucial for the interaction between blood and the receptor *Vmn2r88*. Furthermore, the authors show that hemoglobin acts as a chemosensory signal to the brain that has an important behavioral function. They do so by showing that hemoglobin exposure results in “exploratory” behaviors in lactating mothers.

Even though the G protein-coupled receptor *Vmn2r88* was recently identified as important for hemoglobin sensing in mice (Isogai et al., *Cell* 2018), the authors make additional important contributions that make the first part of the paper particularly impressive. Specifically, the authors show that Gly17 on hemoglobin is a crucial site for receptor interactions. This finding might also provide an evolutionary perspective on blood sensing and is crucial for understanding the interaction between hemoglobin and the *Vmn2r88* receptor. Another central addition to the Isogai et al., paper is the second part of this study, which identifies a behavioral output following the olfactory sensing of hemoglobin, and identifies potential downstream neuronal circuits that are important for executing the behavioral outputs. This is an important new discovery as in Isogai et al., the authors could not identify a major behavioral affect when mutating the *Vmn2r88* receptor. Taking these new discoveries into

consideration, this paper can be suitable for publication in Nature Communications if some further work would be added to strengthen the second part of this study (see below).

We appreciate the reviewer's careful read and all the valuable and constructive comments.

Major concerns:

1. Mothers and non-parental mice. While most of the major new findings in the second part of the paper are performed in lactating females, the optogenetics and experiments of Figs. 1-3 are performed in naïve males. At the very least, the optogenetic experiments should be performed in lactating females as well. Furthermore, it would be helpful to know if the Hb fos responses are comparable in the VNO & AOB of mothers vs. naïve females. Such data could add mechanistic insights to the changes in responses to hemoglobin observed between naïve females and mothers in downstream brain regions. Other work on the accessory olfactory system seem to support this line of inquiry (Dey et al., Cell 2015, Xu et al., Neuron 2016).

We thank the reviewer for the comment that is important to interpret our results. We performed optogenetic experiments with mothers (**Revise figure 6 below, new Figure 7, and Supplementary figure 10** in the revised manuscript). We injected the virus expressing ChR2 in a Cre-dependent manner and GFP-virus as a control into the VMHd of *SF1-Cre* female mice. Then breeding with male mice started during viral incubation. A couple of days after their childbirth, we performed optogenetic stimulated assays. Weak light stimulation (0.01 mW) caused significant digging behavior enhancement only in ChR2-virus injected group (**Revise figure 6d**). With 0.03 mW stimulation, there was a trend of increase in the digging behavior (**Revise figure 6e**). On the other hand, assays with stronger light stimulation (1 mW) caused freezing, jump, and dash away (**Revise figure 6f and 6g**), consistent with that report that SF1-positive cells in the VMHd are responsible for defensive behaviors (Kunwar et al.⁴).

We also performed *Vmn2r88* ISH and pS6 immunostaining with sections of the VNO from both lactating and virgin females and *c-Fos* ISH with sections including the AOB from lactating females (**Revise figure 7, new Supplementary figure 4** in the revised manuscript). The results suggest that there was no difference in the response to hemoglobin in peripheral receptor-expressing vomeronasal cells and neurons in the AOB between mothers and virgin females, suggesting that the changes in response occur in downstream brain regions. Although there are situations that the response of VSNs can be changed in a state- or experience-dependent fashion (Dey et al., Xu et al.^{9,10}), patterns of neural response in the MeA or downstream regions can be important to hemoglobin-dependent behavioral outputs. This idea is indeed consistent with the recent study about a protein pheromone, darcin, that shows that the responsible circuit from the AOB to MeA can be indispensable to pheromone-dependent specific outputs by integration of pheromonal information and other factors such as internal state (Demir et al.¹¹).

Revise figure 6

Weak activation of SF1-positive cells in the VMHd recaptures digging behavior in lactating females.

a Schematic image and illustration of the setup and timeline for optogenetic activation of *SF1* expressing neurons in the VMHd of lactating females. AAV-DIO-ChR2 or AAV-DIO-GFP are injected into SF1-positive cells in the VMHd and optic fibers are implanted above the target region. Image adapted from The mouse

brain in stereotaxic coordinates (Academic Press, 2007)⁸. **b** A representative coronal section showing ChR2 expression (eYFP positive cells shown in green) in the VMHd. Scale bar, 500 μ m. **c-e** Quantification of digging, rearing, and self-grooming behaviors, with or without weak light stimulation (**c** 0 mW, **d** 0.01 mW, **e** 0.03 mW). Error bars, S.E.M. * $p < 0.05$ by the Wilcoxon signed-rank test, $n = 5$. **f** Quantification of digging, rearing, self-grooming behaviors, and jumping, with or without stronger light stimulation (1 mW). Error bars, S.E.M. n.s. * $p < 0.05$ by the Wilcoxon signed-rank test, $n = 5$. **g** A pie chart showing the distribution of behavior outputs by lactating females stimulated by stronger light (1 mW).

Revise figure 7

Hemoglobin activates Vmn2r88 expressing cells in the VNO and neurons in the AOB in both lactating and virgin females.

a *Vmn2r88* ISH and pS6 immunostaining of VNO sections from mother and virgin female mice exposed to hemoglobin (Hb) or distilled water. Scale bar, 50 μ m. **b** Quantification of visualized neurons in the VNO. The number of pS6-(green), *Vmn2r88*-(magenta), and double-positive cells (yellow) per VNO section. 9 sections from each of 3 animals were quantified. **c** Representative immunohistochemical images (left: Mother-control, middle: Mother-300 μ g Hb, right: Virgin-300 μ g Hb) of ISH with *c-Fos* probe in the sections including the AOB. Scale bar, 100 μ m. Abbreviations: Gl, Glomerular layer; M/T, Mitral/tufted cell layer. **d** Quantification of *c-Fos*-positive neurons in the Glomerular layer (Gl) and Mitral/tufted cell layer (M/T) of the AOB. The number of sections counted to determine the number of *c-Fos*-positive neurons in each brain area of each animal was 6. Error bars, S.E.M. $n = 3$. Abbreviations: ant, anterior; post, posterior.

2. Behavioral responses to hemoglobin. In order to identify how hemoglobin affects behaviors through neuronal activity, the authors show: 1. That exposure to hemoglobin results in elevation of *cfos* expression in VMHdm of mothers. 2. That hemoglobin exposure results in digging and rearing behaviors. 3. That artificial activation of SF1+ neurons in VMHdm can result in behaviors such as digging and rearing in naïve males. However, this does not clarify the question of whether or not hemoglobin actually elicits digging&rearing behaviors through SF1+ neurons in VMHdm in mothers. In order to connect these observations, the authors should use a loss of function approach. Ideally, the authors should reversibly silence SF1+ neuronal activity in VMHdm and test whether the behavioral responses to hemoglobin are affected. This can potentially show that SF1+ neurons in VMHdm are necessary for executing rearing&digging behaviors upon sensing hemoglobin.

Thank you so much for important comments about the manipulation assays of hypothalamic regions. We performed the loss-of-function experiment using virus encoding DREADD-Gi to silence neural activities of the SF1-positive cells in the VMHd of lactating *SF1-Cre* female mice (**Revise figure 4 below**, added into **new Figure 7** in the revised manuscript). Virus injection into SF1-positive cells in the VMHd was performed before breeding. After waiting for sufficient viral infection and their childbirth, CNO or saline injection into the lactating female mice was performed 60 minutes before hemoglobin exposure and at the same time, their pups were removed from the cage. Behavioral recordings just after hemoglobin stimulation showed that neural silencing elicited a significant decrease in the digging behavior duration (**Revise figure 4c below**). These results suggest that there is the necessity of SF1 expressing neurons in the VMHd for hemoglobin-dependent digging enhancement.

Revise figure 4

SF1-positive neurons in the VMHd are necessary for hemoglobin dependent digging behavior.

a Schematic illustration of the animal setup and timeline for pharmacological inhibition of *SF1* expressing neurons in the VMHd. AAV-DIO-hM4Di-mCherry is injected into SF1-positive cells in the VMHd. Image adapted from The mouse brain in stereotaxic coordinates (Academic Press, 2007)⁸. **b** A representative coronal section showing DREADD-Gi expression (mCherry positive cells shown in red) in the VMHd. Scale bar, 500 μ m. **c** Quantification of the digging and rearing total time duration (sec) of Hb stimulated *SF1-Cre* lactating mothers with pre-saline i.p. injection and CNO i.p. injection. $n = 4$. Error bars, S.E.M. * $p < 0.05$ by the Wilcoxon signed-rank test.

3. “Risk assessment” vs. “exploratory” behavior. The authors use the term “exploratory behaviors” when they describe the behavioral outputs observed in response to hemoglobin (in mothers), based on negative state (e.g., cort levels). However, it can be argued that the authors could not detect anxiety markers because the anxiety levels are too low to produce detectable increases. Furthermore, the behavioral data that are presented in the paper may be more accurately described as risk assessment, a mild form of defensive behavior or what Fanselow and Zhuravka described as an “early defensive mode” (Fanselow and Zhuravka, Behavioral Processes 2019). Importantly, it has been demonstrated earlier that the VMHdm and specifically SF1+ neurons in VMHdm are important in processing defensive behaviors including rearing (Kunwar et al., eLIFE 2015, Wang et al., Neuron 2015). In addition, digging can be considered as a form of a risk assessment (for example Gozzi et al., Neuron 2010). The authors should therefore consider switching/adding the term “risk assessment” in order to be more consistent with the literature (Gross and Canteras, NRN 2012, Ribeiro-Barbosa et al., Neuroscience & Biobehavioral Reviews 2005). Interestingly, the authors show using an innate

fear inducing volatile odor (2MT, fig.5f), that mice transit from an immobility-freezing behavior elicited by high doses of 2MT (freezing due to immediate danger) to digging behavior ("reduce the likelihood of encountering danger" Fanselow and Zhuravka, Behavioral Processes 2019). Since both low doses of 2MT and hemoglobin elicit similar risk assessment-exploratory behaviors, it would be informative to examine whether a lower dose 2-MT activates the same SF1+ VMHdm neurons as Hb in mothers (e.g., by fos-catFISH or single-cell imaging). This is especially important since TMT, a compound related to 2-MT, has previously been shown not to activate VMHdm (Perez-Gomez et al., 2015), in contrast to other predator odors.

We agree with the reviewer’s comment. We changed the relevant descriptions in the manuscript to ‘exploratory and/or risk assessment behavior’. Regarding the VMHd activation, we did not see a significant increase in *c-Fos* positive neurons upon stimulation with a lower dose of 2-MT as was the same for TMT (**Revise figure 8 below**) (the number of *c-Fos*+ cells, mineral oil; 122.8 ± 7.3 ($n = 4$), low 2MT in mineral oil; 124.4 ± 23.1 ($n = 5$), suggesting that exposure to 2-MT and hemoglobin elicit a similar behavior but they possess different biological meaning.

Revise figure 8

Low concentration 2MT does not activate neurons in the VMHd.

a Representative coronal sections from C57BL/6 lactating females stimulated with low concentration 2MT (10000-fold dilution) or mineral oil control showing *c-Fos* expression in the VMHd. Scale bar, 200 μ m. **b** Quantification of total *c-Fos* positive cells in the VMHd of lactating mothers. $n = 4-5$. Error bars, S.E.M.

Minor concerns:

- It has been demonstrated that outputs from VMHdm to AHN are important for avoidance behaviors, and that VMHdm to dPAG is important for immobility (Wang et al., Neuron 2015). The authors could address these findings made by previous works through

examining cFOS expression in the AHN of mothers (and if needed, change figure 7f accordingly).

- In several places, the authors measure cfos activity "Per region". This is not clear. Did the authors collect the whole region? Did the authors randomly apply a single brain slice from each region? Since many of the observations described here rely on this analysis, this should be explained in more detail and include exact coordinates.

We examined hemoglobin-induced *c-Fos* expression in the anterior hypothalamic nucleus (AHN) of lactating females (**Revise figure 9 below**). There was no significant increase in *c-Fos* expression in the AHN (the number of *c-Fos*⁺ cells, control; 37.8 ± 7.8 ($n = 4$), hemoglobin; 60.3 ± 14.6 ($n = 5$), $p=0.05$ by the Wilcoxon signed-rank test).

We apologize that our previous explanation about the quantification of staining results was not sufficient. In our staining experiments, we collected multiple sections from each region to cover entire populations (eg; in the VMHd we counted activated cells in the 5 coronal sections from anterior to posterior). As the reviewer suggested, we added the number of sections we counted per region in the legends of **Figure 4** and **Supplementary figure 5** and described cell counting into the method section of **Histochemistry**.

Revise figure 9

Hemoglobin does not activate neurons in the anterior hypothalamic nucleus.

a Representative coronal sections from C57BL/6 lactating females stimulated with Hb or vehicle control showing *c-Fos* expression in the anterior hypothalamic nucleus. **b** Quantification of total *c-Fos* positive cells in the anterior hypothalamic nucleus of lactating mothers. $n = 4-5$. Error bars, S.E.M.

- Why are there responses in the VNO of naïve males and females if they have no behavioral relevance? The authors should clarify this in the text.

Our results from c-Fos analysis and manipulation assays suggest that a responsible circuit of hemoglobin signal in lactating females seems to be VNO-AOB-MeA-VMHd (SF1-positive cells)-PAGd. As it was pointed out, both male mice and virgin females showed VNO activity towards hemoglobin (**Figure 3** and **Revise figure 7a-b above**), but there was no clear induction of *c-Fos* expression in higher brain regions such as the VMHd and PAG (**Figure 4**) and no behavioral outputs in male and virgin females (**Figure 5d**). It is possible that this difference was caused at downstream regions that are affected by the internal state such as lactating. The results are somewhat consistent with those in the recent paper about non-volatile protein pheromone, darcin, suggesting that the circuit from the AOB to MeA can be responsible for the integration of pheromonal information, whereas the transmission process to downstream brain regions is affected by other factors such as the internal state (Demir et al.¹¹). We added discussion in the revised manuscript (**page 9 line 20-27**).

- What is the role for the main olfactory bulb in hemoglobin sensing in mothers? This is important especially since it has been shown that there is functional plasticity to behaviorally relevant odors in the main olfactory bulb of mothers (Vinograd et al., Cell Reports 2017).

As the reviewer mentioned, plasticity in the main olfactory system is relevant to odor-driven behaviors in mothers. However, as for specific behaviors observed in hemoglobin-stimulated mothers, the genetic deletion of *Vmn2r88* impaired these outputs (**Figure 5c and 6c**), suggesting that only the vomeronasal olfactory system is involved in evoking hemoglobin-dependent behaviors and the main olfactory system seems to have a minor role in this context in lactating females. Nonetheless, a possible involvement of volatile odor signal through the main olfactory system remains to be elucidated.

- Figure 6 b and 6 c. What is the difference between the control groups in figure 6b (rearing time) and figure 6c? Similarly, what is the difference between the hb condition in figure 6b and hb +/+ condition in figure 6c? If these are different animals for the same conditions the authors should group them together to increase the statistical power.

In figure 6b, we used pure C57BL/6 mice purchased from Japan SLC or Japan CLEA to perform the open field assay. And then the result showing hemoglobin-dependent rearing enhancement led us to perform the additional assay with *Vmn2r88* mutant mice to confirm the output could be controlled by hemoglobin receptor; *Vmn2r88*. *Vmn2r88* knock-out mice are also C57BL/6 background but its origin was different from C57BL/6 mice from the vendors, so we made the control group for the assay using *Vmn2r88*^{-/-}. Siblings of *Vmn2r88* knock-out females (*Vmn2r88*^{+/+}) were used in figure 6c.

- Figure 1, individual mice. While in all figures the variance can be estimated by explicitly showing the data for individual subjects on top of the bar graphs, it will be informative to add this in figure 1 as well (and not just error bars).

Thanks for pointing this out. We added the data of individual subjects in **Figure 1a, b, c, d, e, h**, and **Supplementary figure 2a**.

- In supplemental figure 7, mice spent similar time in a 2-chamber assay where hemoglobin was compared to the control. However, several concerns arise from this interpretation made by the authors. Most importantly, it seems that the mice tested in the assay were male mice and not mothers. This is a surprising choice for this assay, considering the fact that males did not show a significant behavioral difference in other assays, and other works imply that hemoglobin might be attractive to males (Isogai et al., Cell 2018).

In our 2-chamber assay (now in **new Supplementary figure 8** in the revised manuscript), we used lactating females as test mice. We apologize that our explanation in the figure and text was not clear. We modified the figure legends and text to clearly mention that lactating mothers were used.

In summary this is a strong submission and our comments should be taken as constructive remarks to improve the paper and eliminate confusion and ambiguity for the readers. We congratulate the authors on their beautiful and impressive work.

Thank you very much for all the constructive comments and suggestions. We greatly appreciated all comments that made this study more convincing.

Reviewer #3 (Remarks to the Author):

In this paper, Osakada and colleagues – stimulated by initial observations that blood activates the VNO – identify the specific blood ligand responsible for this activation (beta-globin) and its corresponding VNO receptor (Vmn2r88). Using Vmn2r88-knockout mice, they demonstrate that smelling blood causes exploratory behavior (rearing and digging) in a receptor-dependent manner, and that this response is elicited in lactating mothers, but not virgin females or males. The behavioral response is attributed to the specific activation of dorsal PAG, MeA and dorsal VMH (dVMH) in lactating mothers, and was recapitulated by optogenetic activation of SF1 neurons in dVMH. On the basis of these data, the authors claim that the smell of blood activates a specific behavioral response that is dependent on an AOB-MeApy-VMHd-PAGd pathway and unique to lactating mothers.

Overall, the experiments identifying beta-globin and Vmn2r88 (Figs 1 and 3) in this paper are elegant and convincing, and consistent with a recent paper (Isogai, 2018) that also found beta-globin and Vmn2r88 to mediate the sensing of blood VNO. The subsequent neural and behavioral analysis, however, could be improved to rule out alternative explanations for the

authors claims. In particular, I am concerned that the data may reflect a general response to VNO activation that could include a much larger category of ligands than just beta-globin.

For example, the authors show that 2MT also elicits digging behavior when presented at very low concentration. In addition to this overall feedback, I have several concerns and questions that will clarify this manuscript:

1. To clarify the specific effects of hemoglobin, it would be helpful to see the same behavioral experiments repeated for other ligands, including some that would likely covary with hemoglobin (e.g. other infant-associated ligands), and others that are independent.

Thank you very much for careful read and important suggestions. To show the specificity of hemoglobin itself as a vomeronasal ligand for digging behavior, we added some control groups. Here, we used ESP1, one of the peptide pheromones received by another specific vomeronasal receptor (Vmn2r116, also known as V2Rp5) as a negative control (Haga et al.¹²). ESP1 did not enhance digging behavior (**Revise figure 10b below**) nor rearing behavior (**Revise figure 2b**) in lactating mothers. We also added a fresh blood-stimulated group as a positive control into experiments as well (digging assay: **Revise figure 10 below**, pup retrieval assay: **Revise figure 12 below**). These results support that there is a specificity of hemoglobin's effect on digging behavior in lactating females. In addition, we performed the open field assay with ESP1 and 2MT stimulations (**Revise figure 2**). The results that both ESP1 and 2MT stimulation did not increase rearing behavior suggested that enhancement of rearing behavior in the open field assay is specifically derived by hemoglobin.

Revise figure 10

Hemoglobin and fresh blood but not ESP1 enhanced digging behavior of lactating female mice

a Timeline for digging assay using lactating female mice and with cotton exposure swabbed with either ESP1 (20 μ g), hemoglobin (Hb, 300 μ g), fresh blood (30 μ L), or control buffer. Pups were removed from the

mother's cage 60 min prior to cotton exposure and behavior recording. **b** Quantification of the total digging time duration (sec) of lactating mothers with prior cotton exposure. $n = 6-15$. Error bars, S.E.M. $*p < 0.05$ by the Wilcoxon signed-rank test.

Revise figure 2

Hemoglobin but not ESP1 and 2MT enhances rearing, a type of exploratory behavior, in lactating mothers.

a Schematic illustration of the timeline of the open field assay with cotton pre-exposure. Cotton exposure was performed in their home cage (with their pups). **b** Quantification of total distance, total center time, moving speed, and rearing time duration of lactating mothers, pre-stimulated with control buffer-, ESP1-, 2MT- (low 2MT: 10000-fold dilution, high 2MT: 10-fold dilution) or Hb-cotton swabs in the open field assay for 10 minutes. $n = 8$ for control, $n = 7$ for Hb, and $n = 5$ for ESP1, low 2MT, and high 2MT. Error bars, S.E.M. $*p < 0.05$ by Steel-test. **c** Quantification of rearing time duration of *Vmn2r88* mutant lactating mothers, pre-stimulated with control buffer- or Hb-cotton swabs. $n = 6-10$. Error bars, S.E.M. $*p < 0.05$ by the Steel-Dwass test.

2. Fig 7 shows the result of optogenetic stimulation of SF1 neurons in dVMH. SF1 neurons are allegedly involved in fear responses and comprise the majority of dVMH cells. Interpreting this result requires knowing whether blood activates these SF1 neurons.

We appreciate the reviewer's valuable comment regarding SF1-positive neurons. We did two-color *in situ* hybridization of an immediate early gene, *c-Fos*, and *SF1* mRNA probes with sections from lactating mice with prior blood stimulation. The staining demonstrated that hemoglobin could activate *SF1*-positive cells in the VMHd (**Revise figure 3 below**), and the number of *SF1* and *c-Fos* double-positive cells was larger than that of *c-Fos* and *SF1*-negative cells (the number of *SF1*-negative and Hb-dependent *c-Fos*⁺ cells; 37.8 ± 7.8 , the number of *SF1*-positive and Hb-dependent *c-Fos*⁺ cells; 60.3 ± 14.6 , both shown in black in **Revise figure 3b**). The SF1, whose expression is mostly limited in the VMHd (McClellan et al.³), appears to be a good molecular marker to manipulate the cells in the VMHd as shown in most of the studies focusing on the VMHd (Kunwar et al., Wang et al., Zhang et al.⁴⁻⁶)

Revise figure 3

Hemoglobin activates SF1 expressing neurons, a specific molecule marker for the sub-region, in the VMHd.

a Images from dual-color ISH staining of sections including the VMH from hemoglobin (Hb) or distilled water (control) -stimulated C57BL/6 mother mice labeled with the *SF1* cRNA probe (green) and *c-Fos* cRNA probe (magenta). Scale bar, 100 μ m. **b** Quantification of *c-Fos*-positive neurons overlapping with or without *SF1* in the VMHd. The number of sections counted to determine the number of *c-Fos*-positive neurons in each brain area was 6. Error bars, S.E.M. $n = 4$. * $p < 0.05$ by unpaired two-sided Student's *t*-test.

3. Figure 2d,e is used to show that the H78N mutation in beta-globin does not inhibit VNO activation. However, data are only shown for two mutant proteins and a no-odor control, but not for the WT protein. Thus it cannot be determined from the figure whether H78N leads to a reduction of VNO activation compared to the WT protein.

We added the data of BALB/c male hemoglobin (BALB/c Hb) into figure 2e (and **Revise figure 11 below, new Figure 2e** in the revised manuscript). The number of c-Fos positive cells upon stimulation with H78N was as many as those with the WT protein.

Revise figure 11

G17A in hemoglobin is a crucial amino acid residue for the vomeronasal activity of hemoglobin in mice

Quantification of total c-Fos positive cells in the M/T layer of the AOB of male mice. $n = 3-9$. Error bars, S.E.M. * $p < 0.05$ by the Steel-Dwass test.

4. Is the ability of VNO neurons to detect hemoglobin state dependent (like the fos activity reported in the brain)?

We performed additional staining with the VNO sections (**Revise figure 7a and 7b below**, added into **new Supplementary figure 4** in the revised manuscript), showing that VNO neurons in both lactating mothers and virgin females were activated by hemoglobin and thus that there is no state-dependent detection at the VNO level. We made some discussion regarding this observation in **page 9 line 20-27**.

Revise figure 7a and b

Hemoglobin activates *Vmn2r88* expressing cells in the VNO in both virgin females and lactating mothers.

a *Vmn2r88* ISH and pS6 immunostaining of VNO sections from mother and virgin female mice exposed to hemoglobin (Hb) or distilled water (control). Scale bar, 50 μm . **b** Quantification of visualized neurons in the VNO. 9 sections from each of 3 animals were quantified. The *x*-axis shows that the number of pS6-(green), *Vmn2r88*-(magenta), and double-(yellow) positive cells per VNO section.

5. If blood is placed on a pup, how does the mother behaviorally react?

We conducted a new pup retrieval assay with a slight modification. We painted fresh blood, hemoglobin, and control vehicle directly onto the back of pups just before the retrieval assay. The result showed that not only hemoglobin but fresh blood painting elicited delay in pup retrieval and digging enhancement of lactating females (**Revise figure 12 below, new Supplementary figure 7** in the revised manuscript).

Revise figure 12

Pup retrieval assay with C57BL/6 lactating female mice.

a Timeline for pup retrieval assay using lactating female mice and pups painted either hemoglobin (300 μg Hb), fresh blood (30 μL), or control buffer. Pups were removed from the female's cage 30 min prior to behavior recording. **b** Combined percentage of pups with Hb, fresh blood, or control buffer-painted on their back (out of three) retrieved by an animal group as a function of time. **c** Quantification of the digging time duration of lactating mothers in pup retrieval assays. $n = 5$ for control, $n = 7$ for Hb, and $n = 6$ for fresh blood. Error bars, S.E.M. * $p < 0.05$ by the Steel-Dwass test.

[REFERENCES]

1. Tsunoda, M. *et al.* Identification of an Intra- and Inter-specific Tear Protein Signal in Rodents. *Current Biology* **28**, 1213-1223.e6 (2018).
2. Osakada, T. *et al.* Sexual rejection in female mice via a vomeronasal receptor-triggered limbic circuit. *Nature Communications* **9**, 4463 (2018).
3. McClellan, K. M., Parker, K. L. & Tobet, S. Development of the ventromedial nucleus of the hypothalamus. *Frontiers in Neuroendocrinology* **27**, 193–209 (2006).
4. Kunwar, P. S. *et al.* Ventromedial hypothalamic neurons control a defensive emotion state. *eLife* **2015**, 1–30 (2015).
5. Wang, L., Chen, I. Z. & Lin, D. Collateral Pathways from the Ventromedial Hypothalamus Mediate Defensive Behaviors. *Neuron* **85**, 1344–1358 (2015).
6. Zhang, J., Chen, D., Sweeney, P. & Yang, Y. An excitatory ventromedial hypothalamus to paraventricular thalamus circuit that suppresses food intake. *Nature Communications* **11**, (2020).
7. Dulac, C. & Wagner, S. Genetic Analysis of Brain Circuits Underlying Pheromone Signaling. *Annu. Rev. Genet* **40**, 449–67 (2006).
8. Franklin, K. & Paxinos, G. *The mouse brain in stereotaxic coordinates. The Spinal Cord: A Christopher and Dana Reeve Foundation Text and Atlas* (2008). doi:10.1016/B978-0-12-374247-6.50004-3
9. Dey, S. *et al.* Cyclic regulation of sensory perception by a female hormone alters behavior. *Cell* **161**, 1334–1344 (2015).
10. Xu, P. S., Lee, D. & Holy, T. E. Experience-Dependent Plasticity Drives Individual Differences in Pheromone-Sensing Neurons. *Neuron* **91**, 878–892 (2016).
11. Demir, E. *et al.* The pheromone darcin drives a circuit for innate and reinforced behaviours. *Nature* **578**, 137–141 (2020).
12. Haga, S. *et al.* The male mouse pheromone ESP1 enhances female sexual receptive behaviour through a specific vomeronasal receptor. *Nature* **466**, 118–22 (2010).

Reviewers' Comments:

Reviewer #1:

Remarks to the Author:

The revised manuscript is much improved with new data and clarification. However, I have one further concern about the new data. The loss of function experiment implicating the SF1 neurons in hemoglobin-evoked behavior is great (Figure 7c), but it is missing an important control. The authors express the DREADD, hM4D-mCherry, in these neurons and show the administration of CNO eliminates behaviors evoked by hemoglobin. However, a control group of mice expressing mCherry needs to be included. What if CNO alone causes the loss of the exploratory behaviors? This seems particularly important given the increasing reports of off target effects of CNO.

Also with regard to CNO concentration, it appears that the authors use a concentration that is an order of magnitude lower than what is typical (~1ug/kg vs 1mg/kg). Is this really correct?

Reviewer #2:

Remarks to the Author:

This elegant study identifies a specific receptor in the vomeronasal organ (VNO) of mice, Vmn2r88, that is sensitive to blood. The authors further demonstrate that the amino acid Gly17 in the hemoglobin protein is crucial for the interaction between blood and the receptor Vmn2r88. This finding might also provide an evolutionary perspective on blood sensing and is crucial for understanding the interaction between hemoglobin and the Vmn2r88 receptor.

Furthermore, the authors show that hemoglobin acts as a chemosensory signal to the brain, resulting in exploratory/risk assessment behaviors only in lactating mothers. They also identify downstream neuronal circuits important for this state-dependent execution of behavior.

Taking these discoveries into consideration, we believe that this paper is suitable for publication in Nature Communications. We congratulate the authors on their impressive work.

Reviewer #3:

Remarks to the Author:

Our original review focused on some ambiguity in the interpretation of the perturbation and behavior experiments: especially relating to (1) the specificity of the response to blood/Hb; (2) the validity of optogenetic stimulation for recapitulating ligand-induced activity.

The authors responded to these concerns by (1) including results for additional ligands (EPS1 and 2MT); (2) showing that Hb-activated neurons in dVMH are enriched among the targeted (SF1+) neural population. The revisions are a substantial improvement over the manuscript as submitted, and give confidence in the conclusions.

One limitation of the current data is the difficulty of interpreting the function of increased digging and rearing behavior. The authors' suggestion that it represents an exploratory or risk-assessment state is reasonable given the limited insight afforded by open field behavior alone, but it would be useful for the discussion to make stronger contact with the ethology of wild mice in order to the possible behavior states that Hb elicits.

Responses to reviewers' comments

We greatly thank all reviewers for the enthusiasm about our study, and for their constructive comments. We have performed one additional experiment and some modifications in response to the important comments from the reviewers, which have resulted in significant improvement of the paper.

Additional experiments performed:

- **Adding control experiments of pharmacogenetic loss-of-function assays in SF1-expressing neurons in the VMHd. (Supplementary Figure 10, Revise figure 1d and 1e)**

Point-by-point responses:

Reviewer #1 (Remarks to the Author):

The revised manuscript is much improved with new data and clarification. However, I have one further concern about the new data. The loss of function experiment implicating the SF1 neurons in hemoglobin-evoked behavior is great (Figure 7c), but it is missing an important control. The authors express the DREADD, hM4D-mCherry, in these neurons and show the administration of CNO eliminates behaviors evoked by hemoglobin. However, a control group of mice expressing mCherry needs to be included. What if CNO alone causes the loss of the exploratory behaviors? This seems particularly important given the increasing reports of off target effects of CNO.

RESPONSE: We thank the reviewer for the comment. We performed additional pharmacogenetic loss-of-function assays in the VMHd of wild type lactating females to make up for the missing control. Here we used the same experimental conditions originally mentioned in our method section, not using *SF1-Cre* transgenic female mice but wild type C57BL/6 female mice. In the VMHd of virus injected wild type female mice, there was no viral gene expression (no mCherry expression that represents the expression of DREADD-Gi encoded in the virus) in the VMHd (**Revise figure 1d**) and there was no significant decrease in total digging duration after CNO administration comparing with that of saline (**Revise figure 1e**). This data supports our original results that showed a significant decrease in digging behavior after CNO administration in *SF1-Cre* mothers infected with DREADD-Gi encoding virus (**Revise figure 1b and 1c**). Taken together, this behavioral change was not because of administration of CNO itself but of silencing the activity of specific neurons in lactating female mice. We put this new result in **Supplementary Figure 10** of our updated manuscript.

Revise figure 1

SF1-positive neurons in the VMHd are necessary for hemoglobin-dependent digging behavior.

a Schematic illustration of the animal setup and timeline for pharmacogenetic inhibition of *SF1*-expressing neurons in the VMHd. AAV-*DIO-hM4Di-mCherry* is injected into *SF1*-positive cells in the VMHd. In control group, AAV-*DIO-hM4Di-mCherry* is injected into the VMHd of wild type (WT) C57BL/6 female mice. Image adapted from The mouse brain in stereotaxic coordinates (Academic Press, 2007)⁸. **b** A representative coronal section showing DREADD-Gi expression (mCherry-positive cells shown in red) in the VMHd. Scale bar, 500 μ m. **c** Quantification of the total digging duration (sec) of hemoglobin (Hb)-stimulated *SF1-Cre* lactating mothers with pre-saline i.p. injection and CNO i.p. injection. $n = 4$. Error bars, S.E.M. * $p < 0.05$ by the Wilcoxon signed-rank test. **d** A representative coronal section checking DREADD-Gi expression (mCherry-positive cells shown in red if there is viral gene expression) in the VMHd of WT female mice. Scale bar, 500 μ m. **e** Quantification of the total digging duration (sec) of Hb-stimulated WT lactating mothers with pre-saline i.p. injection and CNO i.p. injection. $n = 4$. Error bars, S.E.M.

Also with regard to CNO concentration, it appears that the authors use a concentration that is an order of magnitude lower than what is typical (~1ug/kg vs 1mg/kg). Is this really correct?

RESPONSE: We sincerely apologize that our previous description was not precise. We used 0.1 mg/mL CNO (clozapine-N-oxide, Sigma-Aldrich, C0382), diluted in saline. We have changed the description of the concentration in **the method section (page 17, line 15)**.

Reviewer #2 (Remarks to the Author):

This elegant study identifies a specific receptor in the vomeronasal organ (VNO) of mice, Vmn2r88, that is sensitive to blood. The authors further demonstrate that the amino acid Gly17 in the hemoglobin protein is crucial for the interaction between blood and the receptor Vmn2r88. This finding might also provide an evolutionary perspective on blood sensing and is crucial for understanding the interaction between hemoglobin and the Vmn2r88 receptor.

Furthermore, the authors show that hemoglobin acts as a chemosensory signal to the brain, resulting in exploratory/risk assessment behaviors only in lactating mothers. They also identify downstream neuronal circuits important for this state-dependent execution of behavior.

Taking these discoveries into consideration, we believe that this paper is suitable for publication in Nature Communications. We congratulate the authors on their impressive work.

RESPONSE: We greatly appreciate the reviewer's careful read and all the insightful and constructive comments.

Reviewer #3 (Remarks to the Author):

Our original review focused on some ambiguity in the interpretation of the perturbation and behavior experiments: especially relating to (1) the specificity of the response to blood/Hb; (2) the validity of optogenetic stimulation for recapitulating ligand-induced activity.

The authors responded to these concerns by (1) including results for additional ligands (EPS1 and 2MT); (2) showing that Hb-activated neurons in dVMH are enriched among the targeted (SF1+) neural population. The revisions are a substantial improvement over the manuscript as submitted, and give confidence in the conclusions.

One limitation of the current data is the difficulty of interpreting the function of increased digging and rearing behavior. The authors' suggestion that it represents an exploratory or risk-assessment state is reasonable given the limited insight afforded by open field behavior alone, but it would be useful for the discussion to make stronger contact with the ethology of wild mice in order to the possible behavior states that Hb elicits.

RESPONSE: We thank the reviewer for careful read of the revised manuscript and important suggestions about hemoglobin-dependent output. We added an additional reference suggesting that digging and rearing can be the phenotype of exploratory and/or risk assessment into the discussion (**page 9, line 3**).

Reviewers' Comments:

Reviewer #1:

Remarks to the Author:

The addition of the new data in supplementary Figure 10 satisfies the last of my concerns. The manuscript is now suitable for publication. Congratulations to the authors on a very nice study.